# TEST-TIME ENSEMBLE VIA LINEAR MODE CONNECTIVITY: A PATH TO BETTER ADAPTATION

**Byungjai Kim**[1][*]   **Chanho Ahn**[1]   **Wissam J. Baddar**[1]   **Kikyung Kim**[1]   **Huijin Lee**[1]
**Saehyun Ahn**[1]   **Seungju Han**[1]   **Sungjoo Suh**[1]   **Eunho Yang**[2]
[1]AI Center, Samsung Electronics
[2]Korea Advanced Institute of Science and Technology
{byungjai.kim, chanho.ahn, wisam.baddar, kk87.kim}@samsung.com
{huijin.lee, saehyun.ahn, sj75.han, sungjoo.suh}@samsung.com
eunhoy@mli.kaist.ac.kr

## ABSTRACT

Test-time adaptation updates pretrained models on the fly to handle distribution shifts in test data. While existing research has focused on stable optimization during adaptation, less attention has been given to enhancing model representations for adaptation capability. To address this gap, we propose Test-Time Ensemble (TTE) grounded in the intriguing property of linear mode connectivity. TTE leverages ensemble strategies during adaptation: 1) adaptively averaging the parameter weights of assorted test-time adapted models and 2) incorporating dropout to further promote representation diversity. These strategies encapsulate model diversity into a single model, avoiding computational burden associated with managing multiple models. Besides, we propose a robust knowledge distillation scheme to prevent model collapse, ensuring stable optimization and preserving the ensemble benefits during adaptation. Notably, TTE integrates seamlessly with existing TTA approaches, advancing their adaptation capabilities. In extensive experiments, integration with TTE consistently outperformed baseline models across various challenging scenarios, demonstrating its effectiveness and general applicability.

## 1 INTRODUCTION

As machine learning advances with larger and more complex architectures, the demand for substantial computational resources and extensive datasets rises. This trend has made off-the-shelf pretrained models more valuable than training models from scratch. However, these pretrained models often struggle with data distributions that deviate from their training environments, underscoring the need for effective methods to adapt to diverse distribution shifts and maintain robust performance.

Test-Time Adaptation (TTA) has emerged as an *online* adaptation method for handling distribution shifts in test data. By leveraging off-the-shelf models, TTA enables the adjustment of model parameters to better align with test distributions (Wang et al., 2021). Previous studies have highlighted the importance of maintaining optimization stability to prevent model collapse (Zhang et al., 2022b; Niu et al., 2023; Lim et al., 2023). Addressing more practical environments, some research has tackled stability issues related to dynamic distribution shifts that static models struggle to handle (Gong et al., 2022; Wang et al., 2022; Yuan et al., 2023). However, there has been limited exploration into enhancing model representations during test time to further improve adaptation capabilities.

In parallel, several studies have explored *offline* methods to enhance the adaptability of off-the-shelf models for out-of-distribution. Notably, empirical evidence suggests that fine-tuned models from a pretrained model are 'linearly connected' (Neyshabur et al., 2020), with the fine-tuning operation often approximated by first-order or linear expansions (Jacot et al., 2018; Wortsman et al., 2022; Evci et al., 2022). This property enables straightforward techniques for improving domain generalization. For instance, several approaches have constructed ensembles by averaging the weights of fine-tuned models (Rame et al., 2022; Wortsman et al., 2022), where these ensembles, built from a

---

[*]Corresponding Author

single model, incur no additional computational cost during inference. Recent work has also utilized high-level dropout rates to mitigate shortcut learning and encourage diverse collaboration of existing representations (Zhang & Bottou, 2024). As a result, robust fine-tuning seeks to enhance the representation of off-the-shelf models under linear approximation conditions, demonstrating strong generalization performance across distribution shifts (Zhang et al., 2022a; Zhang & Bottou, 2023).

This paper explores the intersection between TTA and robust fine-tuning, both enhancing model performance on out-of-distribution but differing in their *online* and *offline* operation, respectively. Building on the insight that fine-tuning under linear approximation can streamline adaptation processes, we extend this advantage to TTA. Our preliminary study empirically shows that TTA models can exhibit the linear connectivity. This finding opens the door to incorporating advancements in offline domain generalization into TTA to enrich versatile representations in test time.

We propose Test-Time Ensemble (TTE) that leverages ensemble strategies to dynamically enrich representations during online adaptation. TTE constructs an ensemble network by adaptively averaging the parameter weights of assorted TTA models. Notably, unlike prior methods that require multiple predictions to improve adaptation (Jang et al., 2022; Yuan et al., 2023), this simple weight averaging captures model diversity in a single model, boosting representation quality but reducing the computational burden of multiple model inference. To further promote diverse collaboration among the representations within TTA models, we incorporate dropout, ensuring it does not hinder adaptation or inference. However, existing TTA models often collapse, such as consistently assigning all samples to the same class. Ensemble from such unstable models can lead to performance degradation. To address the instability, we propose de-biased and noise-robust knowledge distillation schemes to stabilize the learning of TTA models within the ensemble. TTE is straightforward to implement and integrates effortlessly with existing TTA methods. By building on the foundational efforts, TTE achieves a significant leap in adaptation performance.

**Author contributions**. 1) We reveal that TTA models exhibit linear mode connectivity, an intriguing insight that simplifies and enhances the adaptation process. 2) Based on this insight, we introduce Test-Time Ensemble (TTE), a novel and computationally efficient approach that not only enriches model representations but also stabilize TTA optimization through de-biased and noise-robust knowledge distillation. 3) TTE integrates effortlessly with existing TTA methods, enhancing adaptation in diverse scenarios and showing potential for applicability to future TTA methods.

## 2 PRELIMINARIES

This section covers the main categories of prior work and preliminary analysis that inspired the proposed TTE. Detailed discussions of related works are included in Appendix A.

### 2.1 TEST-TIME ADAPTATION

Let $f(\theta)$ a pretrained model with parameters $\theta$ for the task $\mathcal{X}_{tr} \to \mathcal{Y}$ where training inputs $x_{tr} \in \mathcal{X}_{tr}$ and labels $y \in \mathcal{Y}$. TTA adapts the model $f$ to learn $\mathcal{X}_{te} \to \mathcal{Y}$ only using out-of-distribution test inputs $x_{te} \in \mathcal{X}_{te}$ in online. The majority of objective functions in TTA are based on entropy minimization (Wang et al., 2021), which aims to reduce uncertainty in the model's predictions, $\hat{y} = f(x_{te}, \theta)$, by adjusting the affine transformation layers of $f$. Specifically, Shannon entropy is adopted as $H(\hat{y}) = -\sum_c p(\hat{y}^c) \log p(\hat{y}^c)$, where $p(\hat{y}^c)$ denotes the probability of class $c$ with the softmax function $p$. Entropy minimization offers a training-independent method that effectively adapts off-the-shelf models for TTA. However, it is prone to model collapse, especially under severe distribution shifts (Gong et al., 2022).

To address challenges in test-time adaptation (TTA), previous approaches have focused on ensuring stable optimization through methods such as selecting reliable samples (Niu et al., 2023; 2022; Yuan et al., 2023; Lee et al., 2024), maintaining prediction consistency (Wang et al., 2022; Zhang et al., 2022b; Chen et al., 2022), manipulating affine transform statistics (Gong et al., 2022; Lim et al., 2023; Zhao et al., 2023), and using robust optimizers (Niu et al., 2023; Gong et al., 2024). In contrast, our work shifts the focus to enhancing adaptation capability, a less explored aspect of TTA research. Some methods have aimed to improve adaptation through dense image augmentation (Yuan et al., 2023; Döbler et al., 2023) or multiple predictions (Jang et al., 2022). Unfortunately, such strategies resulted in heavy computational complexity.

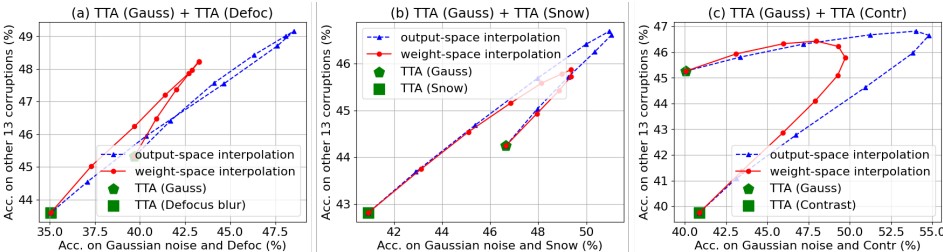

Figure 1: Accuracy trajectories of weight-space and output-space interpolated TTA models in ImageNet-C with 15 corruptions (level 5). The interpolation is performed with adjusting $\alpha$ from 0 to 1 in steps of 0.1. The x-axis shows accuracy for the target corruptions (or distributions) which two TTA models are adapted for and the y-axis for all non-targets. Interpolating model weights (solid line) and interpolating outputs (dashed line) often show similar trajectories.

## 2.2 CAN TEST-TIME ADAPTATION BE APPROXIMATED BY LINEAR EXPANSION?

**Domain generalization under linear approximation**. Several studies provide empirical evidence that fine-tuning a pretrained model can be approximated by a linear expansion when using a dataset much smaller than what is needed to train the network from scratch (Jacot et al., 2018; Maddox et al., 2021). This linear approximation simplifies the process and offers a pathway for applying straightforward strategies to enhance domain generalization. Some researchers have revisited model weight averaging, originally designed for convex problems (Rame et al., 2022; Wortsman et al., 2022), and introduced fine-tuning methods under the premise that fine-tuning primarily leverages existing representations in pretrained models (Evci et al., 2022; Zhang & Bottou, 2024).

We view TTA as a form of domain generalization, where adaptation to currently shifted test samples seeks to improve performance on future samples with potentially different distributions. Building on previous work on fine-tuning with linear expansion, we investigate whether TTA, fine-tuning on domain-shifted test data, can also be approximated by linear expansion.

**Preliminary analysis**. We investigate whether TTA models can be approximated by a linear expansion. Using the setup from Frankle et al. (2020); Wortsman et al. (2022), *Linear Mode Connectivity* in TTA is empirically demonstrated, as described by

$$(1 - \alpha) \cdot \text{Acc}_D(\theta_1) + \alpha \cdot \text{Acc}_D(\theta_2) \leq \text{Acc}_D((1 - \alpha) \cdot \theta_1 + \alpha \cdot \theta_2) \qquad (1)$$

where $\theta_1$ and $\theta_2$ are the weights of two TTA models adapted to different distributions, $\alpha$ is a mixing coefficient in the range $[0, 1]$ and $\text{Acc}_D$ is classification accuracy for a certain distribution $D$. This property suggests that averaging the weights of two models can yield better performance than each model by mimicking output-space ensemble effect. The connection between weight-space and output-space averaging implies that $\theta_1$ and $\theta_2$ can be linearly approximated around the pretrained $\theta_0$, indicating that they are linearly connected (Fort et al., 2020). Neural networks, being non-linear, usually do not benefit from weight interpolation, but linear mode connectivity enables enhanced performance. Appendix B.1 presents theoretical analysis for linear approximation in TTA.

Figure 1 presents the preliminary results for linear mode connectivity, using a visualization method for robustness in distribution shifts (Recht et al., 2019; Taori et al., 2020). Weight-space interpolation can provide higher accuracy both on target and non-target corruptions than single models, following the accuracy trajectories of output-space interpolation. Although trajectory differences grow as the corruption properties become more distinct (noise and blur in Figure 1 (a)), performance improvements are still valid. The results suggest that TTA models can be linearly connected across different distributions, indicating the applicability of domain generalization strategies based on linear approximation. Appendix B.2 describes the details of the preliminary experiments. We also report additional preliminary results under various TTA scenarios in Appendix B.3.

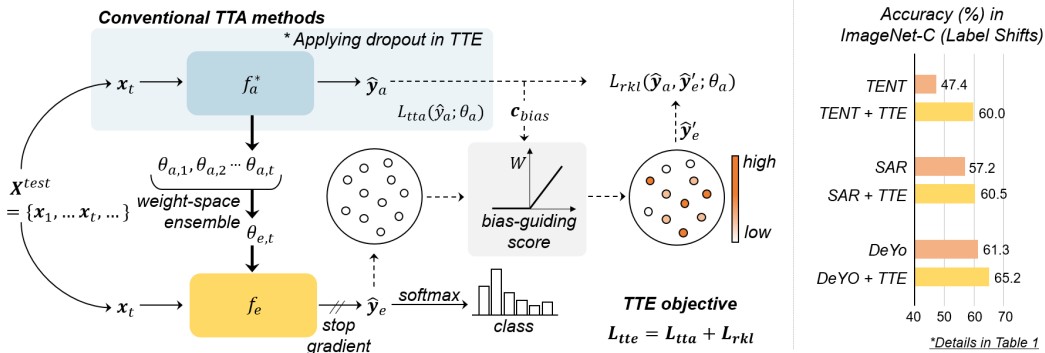

Figure 2: Overall framework of TTE. The ensemble network $f_e$ is constructed from $f_a$ using weight-space and dropout ensembles. Ensemble outputs $\hat{y}_e$ serve as predictions and drive de-biased, noise-robust knowledge distillation to enhance the representation and stability of $f_a$. The procedures integrate seamlessly with existing TTA methods, yielding significant accuracy improvements across three representative methods. Stronger methods achieve even higher performance through TTE.

# 3   TEST-TIME ENSEMBLE

We propose a test-time ensemble (TTE) approach, designed to enhance the adaptation capabilities of existing TTA methods (Figure 2). TTE is built on a teacher-student framework, comprising an adapter network $f_a$ and an ensemble network $f_e$. The proposed TTE involves 1) constructing $f_e$ from orignal TTA models $f_a$ through adaptive ensemble strategies and 2) distilling the knowledge from $f_e$ to $f_a$ to ensure stable optimization. The proposed knowledge distillation maintains linear mode connectivity between $f_a$ and $f_e$ during TTA and also mitigates model collapse, a common issue in TTA processes. Further details are provided in the following sections.

## 3.1   ENSEMBLE STRATEGIES FOR ENHANCING ADAPTATION PERFORMANCE

TTE uses the same input view for both $f_e$ and $f_a$ without augmentation. Instead of multiple predictions with dense augmentations in teacher-student frameworks (Wang et al., 2022; Yuan et al., 2023), we propose computationally efficient ensemble strategies to enhance the representation of $f_e$.

**Adaptive weight-space ensemble**. Our preliminary results in Figure 1 underscore two insights: 1) TTA models even adapted to different distributions would have non-redundant representations and their ensemble are further beneficial for diverse distribution shifts. 2) Weight-space interpolation can emulate the rich representations of ensembles with a single model, eliminating the computational burden of multiple models at test time. In this paper, we propose an online weight-space ensemble approach, where the weights of $f_e$ is iteratively updated via an exponential moving average (EMA) as $\theta_e \leftarrow m\theta_e + (1-m)\theta_a$ with a momentum $m$, where $\theta_e$ and $\theta_a$ denote weights of $f_e$ and $f_a$, respectively. Our preliminary findings suggest that using a wide range of momentum $m$ does not hinder optimization but rather enhances domain adaptation. Thus, we propose an adaptive EMA modulated by the divergence $L_{rkl}$, which measures the probability distance between $f_e$ and $f_a$.

$$m = m_0 \cdot e^{-L_{rkl}/\tau} \tag{2}$$

where $m_0$ is a base momentum and $\tau$ is a temperature controlling sensitivity to the divergence. Eq. 2 promotes a lower momentum to actively construct ensembles when $f_e$ and $f_a$ have different representations. This adaptive scheme is particularly valuable in online TTA, where data distributions are unknown and subject to dynamic shifts. The method for calculating the divergence $L_{rkl}$ is described in Section 3.2.

**Dropout ensemble**. Dropout is traditionally used to introduce noise that hinders optimization, preventing overfitting, and is typically applied in long-range learning a non-linear function from scratch. However, recent studies have shown that applying dropout during fine-tuning create diverse collaboration between existing features Zhang & Bottou (2024), based on the assumption that fine-tuning a pretrained model is a near-linear process that primarily leverages existing representations Evci

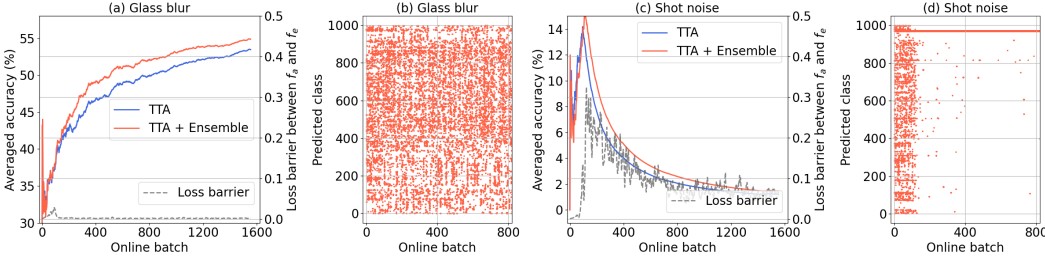

Figure 3: Failure case study. Classification accuracy in applying the proposed ensemble strategies to a conventional TTA approach (Tent, Wang et al. (2021)). The loss barrier between $f_a$ and $f_e$ is measured, as a common metric for assessing linear mode connectivity between them (Eq. 7 in Appendix). The predictions in (b) and (d) present the model outcomes of ensemble $f_e$. Details of this study are included in Appendix D.1

et al. (2022); Jacot et al. (2018). Building on this insight, TTE incorporates dropout in the penultimate layer of $f_a$ during test-time adaptation as $f_a(\cdot; \rho) = h_a(\text{dropout}(g_a(\cdot), \rho))$, where $h_a$ and $g_a$ are the linear head and the backbone of $f_a$, respectively, and $\rho$ is the dropout probability. This dropout encourages model diversity of $f_a$ during exploring distribution shifts in test data, which is then transferred to $f_e$ through weight-space ensemble.

## 3.2  DE-BIASED AND NOISE-ROBUST KNOWLEDGE DISTILLATION

Existing TTA models often collapse, such as consistently assigning all samples to improperly biased classes (Niu et al., 2023). To solve the issues, TTE employs a robust knowledge distillation objective within a teacher-student framework for stable optimization, addressing a key degradation factor in output $\hat{y}_e$: **bias** as a prominent aspect of model collapse.

**De-biased distillation**. Simply constructing ensembles with unstable TTA models can rather degrade performance. Figure 3 illustrates failure cases: while ensemble improves performance under glass blur, it degrades as the TTA model collapses under shot noise, providing long-range biased predictions (Fig. 3(d)). In terms of linear mode connectivity, the loss barrier spikes as the TTA model $f_a$ begins to collapse (Fig. 3(c)). This indicates that the linear connectivity between the TTA model $f_a$ and its ensemble $f_e$ is disrupted on the loss surface, stopping the benefits of the ensemble for adaptation. To solve collapse issues, we introduce a de-biased representation $\hat{y}'_e$ for knowledge distillation to reduce improper bias in $f_a$ and maintain linear mode connectivity. First, we quantify the accumulated bias $c_{bias}$ in $f_a$ by applying EMA to the first-order batch statistics of $\hat{y}_a$, as follows

$$c_{bias} \leftarrow n \cdot c_{bias} + (1 - n) \cdot \frac{1}{M} \sum_{i}^{M} \hat{y}_{a,i} \qquad (3)$$

where $n$ is a momentum constant, $M$ is a batch size and $c_{bias}$ is initialized as all zero. Due to the long-range effects of improper bias, $c_{bias}$ converges into a specific distribution when model collapse manifests. Second, we identify bias-guiding samples from $\hat{y}_e$ by measuring the cosine similarity between $c_{bias}$ and each output $\hat{y}_{e,i}$: $s_i = \frac{c_{bias} \cdot \hat{y}_{e,i}}{\|c_{bias}\| \|\hat{y}_{e,i}\|}$. As $s_i$ increases, $\hat{y}_{e,i}$ is more likely to intensify the bias in $f_a$ during distillation. Third, for bias-guiding samples, $\hat{y}_{e,i}$ is adjusted to construct de-biased representations $\hat{y}'_{e,i}$. The $\hat{y}'_{e,i}$ is generally described with a weight function $w(s) = \max(0, \alpha \cdot s)$ with a scale $\alpha$ as follows

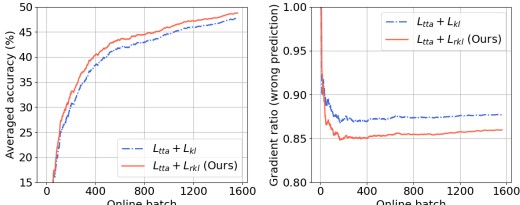

Figure 4: Comparisons of standard and reverse KL divergence with Gaussian noise on ImageNet-C (Level 5), combined with the TTA objective (Tent, Wang et al. (2021)).

$$\hat{y}'_{e,i} = \hat{y}_{e,i} - w(s_i) \cdot c_{bias}. \qquad (4)$$

When $s_i > 0$, $\hat{\boldsymbol{y}}_{e,i}$ is adjusted with $\boldsymbol{c}_{bias}$ to introduce the effects of label smoothing, increasing the probabilistic distance between $\hat{\boldsymbol{y}}_{a,i}$ and $\boldsymbol{c}_{bias}$ during distillation. This adjustment acts as a regularizer for the unsupervised TTA process, helping to prevent model collapse.

**Noise-robust distillation**. Besides, to mitigate the impact of noisy predictions in $\hat{\boldsymbol{y}}_e$, which are used as pseudo-labels for $f_a$ during distillation, we employ reverse KL divergence as the distillation objective. Figure 4 compares standard and reverse KL divergence when they integrate with a conventional TTA approach. Reverse divergence improves accuracy over standard one by penalizing the gradient of incorrect predictions. True labels are used to distinguish correct from incorrect samples, and the gradient ratio is calculated by dividing the gradient of incorrect predictions by that of correct ones. These findings are consistent with results observed in supervised learning with noisy labels (Wang et al., 2019b). The mathematical analysis is included in Appendix D.2.

Consequently, the proposed knowledge distillation objective is mathematically defined as the Kullback-Leibler (KL) divergence between the predictions of $f_a$ and $f_e$ as $L_{rkl}(\hat{\boldsymbol{y}}_a, \hat{\boldsymbol{y}}_e') = KL(\hat{\boldsymbol{y}}_a \| \hat{\boldsymbol{y}}_e')$. This term is also used to dynamically update the momentum value in Eq. 2.

### 3.3 Objective function in TTE

Consequently, the objective function for TTE combines the existing TTA objective $L_{tta}$ with the knowledge distillation objective $L_{rkl}$. Note that $L_{tta}$ is usually based on entropy $H(\hat{\boldsymbol{y}}_a)$ in prior TTA approaches (Wang et al., 2021; Niu et al., 2023; Lee et al., 2024). The combined objective can be interpreted as aiming to reduce the uncertainty of $\hat{\boldsymbol{y}}_a$ while ensuring noise-robust alignment with the de-biased ensemble output from $\hat{\boldsymbol{y}}_e'$, as given by

$$L_{tte}(\hat{\boldsymbol{y}}_a, \hat{\boldsymbol{y}}_e'; \theta_a) = L_{tta}(\hat{\boldsymbol{y}}_a; \theta_a) + L_{rkl}(\hat{\boldsymbol{y}}_a, \hat{\boldsymbol{y}}_e'; \theta_a). \tag{5}$$

To avoid over-tuned hyperparameter configurations, the two objective terms are assigned equal weights. Beyond basic entropy minimization, some prior approaches have proposed reliable sample selection or weighting strategies (Niu et al., 2022; Lee et al., 2024), and robust optimization techniques (Niu et al., 2023). In our framework, we integrate these methods directly into the $L_{tte}$ objective. Algorithms outline how TTE performs with each prior approach in Appendix C.2.

## 4 Experiments

**Datasets and models**. We conducted experiments with four benchmark datasets: **ImageNet-C** (Apache-2.0 License) (Hendrycks & Dietterich, 2019) assesses adaptation performance under 15 types of corruptions at five severity levels, reflecting extreme distribution shifts from the original ImageNet (Deng et al., 2009). All experiments were conducted at level 5, the most severe. **ImageNet-S** (MIT License) (Wang et al., 2019a) and **ImageNet-R** (MIT License) (Hendrycks et al., 2021) evaluated adaptation performance under natural distribution shifts. Unlike ImageNet-C's artificial corruptions, ImageNet-S features sketch-style images for every ImageNet classes, while ImageNet-R includes diverse renditions of 200 ImageNet classes, such as art, cartoons, graffiti, etc. **ImageNet-V2** (MIT License) (Recht et al., 2019) consists of data sampled after a decade of progress on the original ImageNet dataset. It was used to measure performance unaffected by adaptive overfitting, offering a measure of adaptation performance to intrinsic shifts.

We evaluated two types of architectures: Vision Transformer Base (**ViTBase**) and ResNet-50 with Group Normalization (**ResNet50-GN**). Architectures with batch normalization were excluded due to their batch size sensitivity and instability during the TTA process (Niu et al., 2023; Mounsaveng et al., 2024). The models' parameters were initialized using pre-trained weights from the PyTorch Image Models library (Wightman, 2019). For adaptation, the affine parameters of normalization layers in each architecture were trainable. For the ensemble strategies in TTE, the temperature $\tau$ was set to 1.0 with a dropout ratio of 0.9 for ResNet50-GN, and $\tau = 10.0$ with a dropout ratio of 0.4 for ViTBase. The value of $m_0$ was fixed at 1.0 for both ResNet and ViT models. For de-biased knowledge distillation, $n$ was set to 0.99 and $\alpha$ to 3.0. These settings were consistent across all test-time scenarios and baseline methods integrated with TTE to avoid over-tuned hyperparameter configuration. When integrating TTE with existing TTA approaches, we followed the original implementations and hyperparameter settings as specified in their papers to ensure accurate evaluation

Table 1: Integration with previous TTA approaches. Classification accuracy (%) with Label Shifts and Batch Size 1 setups (ImageNet-C, level 5). Underline depicts performance improvement when applying TTE. **Bold** numbers are the best results.

| Label Shifts | Gauss | Shot | Impul | Defoc | Glass | Motion | Zoom | Snow | Frost | Fog | Brit | Contr | Elastic | Pixel | JPEG | Avg |
|---|---|---|---|---|---|---|---|---|---|---|---|---|---|---|---|---|
| ResNet50-GN | 18.0 | 19.8 | 17.9 | 19.8 | 11.4 | 21.4 | 24.9 | 40.4 | 47.3 | 33.6 | 69.3 | 36.3 | 18.6 | 28.4 | 52.3 | 30.6 |
| • Tent | 3.9 | 4.6 | 4.7 | 16.4 | 6.0 | 27.2 | 29.0 | 18.9 | 27.4 | 2.4 | 72.1 | 46.1 | 8.1 | 52.4 | 56.1 | 25.0 |
| • Tent + TTE | 36.2 | 38.4 | 37.3 | 29.6 | 25.8 | 36.1 | 38.6 | 50.2 | 48.8 | 54.9 | 72.1 | 47.7 | 40.1 | 53.0 | 56.4 | 44.4 |
| • SAR | 33.1 | 36.5 | 35.2 | 18.9 | 20.8 | 33.3 | 29.8 | 27.8 | 44.9 | 35.2 | 71.9 | 46.6 | 7.6 | 52.1 | 56.2 | 36.7 |
| • SAR + TTE | 35.9 | 38.4 | 37.3 | 29.7 | 25.3 | 36.2 | 37.4 | 49.8 | 48.0 | 53.2 | 71.9 | 47.6 | 39.1 | 52.8 | 56.2 | 43.9 |
| • DeYo | 28.1 | 44.3 | 42.9 | 23.4 | 16.6 | 41.5 | 6.1 | 52.9 | 52.0 | 20.2 | 73.2 | 53.0 | 37.7 | 60.0 | 59.4 | 40.8 |
| • DeYo + TTE | 43.0 | 45.3 | 44.0 | 34.4 | 33.6 | 43.4 | 46.0 | 55.2 | 53.6 | 61.0 | 73.4 | 54.4 | 51.5 | 61.0 | 60.2 | 50.7 |
| ViTBase | 9.4 | 6.7 | 8.3 | 29.1 | 23.4 | 34.0 | 27.0 | 15.8 | 26.3 | 47.4 | 54.7 | 43.9 | 30.5 | 44.5 | 47.6 | 29.9 |
| • Tent | 30.9 | 1.0 | 23.2 | 54.9 | 53.2 | 58.8 | 54.3 | 13.3 | 12.5 | 69.8 | 76.3 | 66.3 | 59.7 | 69.8 | 66.8 | 47.4 |
| • Tent + TTE | 46.6 | 45.4 | 47.9 | 55.4 | 54.4 | 59.0 | 55.3 | 62.7 | 62.0 | 69.9 | 76.4 | 66.3 | 61.7 | 69.8 | 67.0 | 60.0 |
| • SAR | 46.6 | 29.5 | 48.1 | 55.2 | 54.2 | 59.0 | 54.6 | 58.0 | 44.1 | 69.8 | 76.2 | 66.1 | 60.9 | 69.7 | 66.6 | 57.2 |
| • SAR + TTE | 48.6 | 46.4 | 48.4 | 55.8 | 55.2 | 59.4 | 55.8 | 63.2 | 62.4 | 70.1 | 76.4 | 66.4 | 62.4 | 70.0 | 67.0 | 60.5 |
| • DeYo | 49.1 | 35.9 | 53.6 | 57.6 | 58.6 | 63.8 | 37.5 | 67.9 | 66.0 | 73.1 | 77.9 | 66.5 | 68.6 | 73.5 | 70.1 | 61.3 |
| • DeYo + TTE | 54.0 | 54.7 | 55.2 | 58.9 | 59.7 | 64.5 | 62.1 | 68.3 | 66.7 | 73.9 | 78.0 | 68.3 | 69.3 | 73.8 | 70.3 | 65.2 |

| Batch Size 1 | Gauss | Shot | Impul | Defoc | Glass | Motion | Zoom | Snow | Frost | Fog | Brit | Contr | Elastic | Pixel | JPEG | Avg |
|---|---|---|---|---|---|---|---|---|---|---|---|---|---|---|---|---|
| ResNet50-GN | 18.0 | 19.8 | 17.9 | 19.8 | 11.4 | 21.4 | 24.9 | 40.4 | 47.3 | 33.6 | 69.3 | 36.3 | 18.6 | 28.4 | 52.3 | 30.6 |
| • Tent | 3.1 | 4.1 | 3.7 | 16.6 | 5.2 | 27.2 | 29.0 | 17.7 | 25.1 | 1.9 | 72.0 | 46.2 | 8.1 | 52.7 | 56.3 | 24.6 |
| • Tent + TTE | 41.6 | 43.9 | 42.7 | 33.8 | 31.2 | 41.0 | 44.1 | 53.5 | 52.2 | 59.3 | 73.1 | 51.3 | 47.8 | 57.7 | 58.1 | 48.7 |
| • SAR | 23.4 | 26.5 | 23.9 | 18.4 | 15.1 | 28.6 | 30.3 | 44.4 | 44.8 | 27.4 | 72.3 | 44.7 | 14.6 | 47.0 | 56.1 | 34.5 |
| • SAR + TTE | 25.9 | 28.6 | 26.7 | 23.7 | 17.7 | 30.8 | 32.4 | 48.0 | 46.1 | 42.1 | 72.2 | 45.2 | 34.2 | 47.7 | 56.1 | 38.5 |
| • DeYo | 41.3 | 44.2 | 42.4 | 23.7 | 25.1 | 41.4 | 19.9 | 54.6 | 52.2 | 1.9 | 73.4 | 53.4 | 39.9 | 59.9 | 59.7 | 42.2 |
| • DeYo + TTE | 42.5 | 44.9 | 43.5 | 34.8 | 32.8 | 43.3 | 45.9 | 55.8 | 53.7 | 60.5 | 73.4 | 54.4 | 51.0 | 60.9 | 60.4 | 50.5 |
| ViTBase | 9.4 | 6.7 | 8.3 | 29.1 | 23.4 | 34.0 | 27.0 | 15.8 | 26.3 | 47.4 | 54.7 | 43.9 | 30.5 | 44.5 | 47.6 | 29.9 |
| • Tent | 43.2 | 1.6 | 44.0 | 52.6 | 48.9 | 55.8 | 51.2 | 22.3 | 21.5 | 67.0 | 75.0 | 64.9 | 54.3 | 67.2 | 64.4 | 48.9 |
| • Tent + TTE | 49.2 | 48.8 | 50.1 | 56.2 | 55.8 | 60.2 | 56.9 | 64.3 | 63.6 | 71.4 | 76.9 | 67.0 | 64.1 | 70.7 | 68.1 | 61.6 |
| • SAR | 40.9 | 36.6 | 41.9 | 53.4 | 50.5 | 57.4 | 52.9 | 59.1 | 57.2 | 68.9 | 75.5 | 65.6 | 58.1 | 68.9 | 65.9 | 56.9 |
| • SAR + TTE | 43.6 | 40.4 | 44.3 | 55.2 | 53.1 | 59.2 | 55.4 | 61.5 | 61.9 | 70.7 | 76.7 | 66.7 | 61.9 | 70.2 | 67.3 | 59.2 |
| • DeYo | 53.1 | 51.2 | 54.3 | 58.8 | 59.6 | 64.0 | 37.4 | 68.1 | 66.4 | 73.7 | 78.3 | 68.2 | 68.5 | 73.7 | 70.5 | 63.1 |
| • DeYo + TTE | 53.8 | 54.0 | 54.6 | 59.1 | 59.7 | 64.4 | 62.3 | 68.4 | 66.8 | 73.9 | 78.3 | 68.5 | 69.2 | 73.9 | 70.7 | 65.2 |

of TTE's effects. Comparative methods are introduced in each comparison section. Implementation details are provided in Appendix C.1 and C.2.

## 4.1 COMPARISON STUDY

**Integration with previous TTA approaches**. TTE was applied to three representative TTA approaches to verify its effectiveness and general applicability: Tent (Wang et al., 2021) introduced entropy minimization; SAR (Niu et al., 2023) added sharpness-aware entropy minimization for optimization stability; and DeYO (Lee et al., 2024) introduced object-based sample weighting and selection. We followed the three wild test scenarios from Niu et al. (2023) using ImageNet-C: **Label Shifts** where batches are class-imbalanced with most samples belonging to the same class, **Batch Size 1** where each batch con-

Table 2: Classification accuracy (%) with the Mix Shifts setup (ImageNet-C, level 5). Underline depicts improvement when applying TTE. **Bold** presents the best results.

| Methods | ResNet50-GN | ViTBase |
|---|---|---|
| No Adapt | 30.6 | 29.9 |
| • Tent | 33.1 | 52.3 |
| • Tent+TTE | 38.7 | 57.2 |
| • SAR | 38.1 | 57.1 |
| • SAR+TTE | 39.1 | 57.5 |
| • DeYO | 33.8 | 58.6 |
| • DeYO+TTE | 42.9 | 60.6 |

tains only one sample, testing adaptation with minimal information, and **Mix Shifts** where batches contain samples from various distributions, testing adaptation with multiple shifts simultaneously.

Table 1 presents classification accuracy across 15 distributions in ImageNet-C with the two setups of Label Shifts and Batch Size 1. Table 2 shows classification accuracy for the Mix Shifts setup. The results along with the standard deviations are detailed in Table 14 and 15. Integrating TTE (+TTE) significantly improved accuracy compared to using the original methods alone. Interestingly, while Tent initially performed worse than SAR, Tent+TTE achieved comparable results to SAR+TTE in Label/Mix Shifts and even outperforms SAR+TTE in Batch Size 1 (48.7% vs. 38.5% on ResNet50-GN, and 61.6% vs. 59.2% on ViTBase). This suggests that sharpness-aware optimization in SAR has less benefit for adaptation with small batch size. The DeYO+TTE delivered the best performance across all challenging scenarios, outperforming DeYO. Specifically, it achieved an average accuracy improvement of +9.9% in Label Shifts, +8.3% in Batch Size 1, and +11.1% in Mix Shifts on ResNet50-GN, and +3.9%, +2.1%, and +1.9% on ViTBase. Notably, +TTE remained stable across most setups while previous methods experienced model collapse in certain cases (e.g., shot noise in Label Shifts). The experiments showed that TTE can enhance adaptation performance

Table 3: Continual TTA with non-i.i.d. conditions. Classification accuracy (%) with ImageNet-C (level 5). **Bold** numbers are the best results.

| Methods | Gauss | Defoc | Snow | Contr | Shot | Glass | Frost | Elastic | Impul | Motion | Fog | Pixel | Zoom | Brit | JPEG | Avg |
|---|---|---|---|---|---|---|---|---|---|---|---|---|---|---|---|---|
| ResNet50-GN | 18.0 | 19.8 | 40.4 | 36.3 | 19.8 | 11.4 | 47.3 | 18.6 | 17.9 | 21.4 | 33.6 | 28.4 | 24.9 | 69.3 | 52.3 | 30.6 |
| • Tent | 3.9 | 1.8 | 1.6 | 0.1 | 0.1 | 0.1 | 0.1 | 0.1 | 0.1 | 0.1 | 0.1 | 0.1 | 0.1 | 0.1 | 0.1 | 0.6 |
| • CoTTA | 23.5 | 5.5 | 2.0 | 0.7 | 0.4 | 0.2 | 0.2 | 0.1 | 0.1 | 0.1 | 0.1 | 0.1 | 0.1 | 0.1 | 0.1 | 2.2 |
| • SAR | 33.1 | 16.8 | 44.7 | 44.6 | 42.3 | 18.8 | 45.7 | 37.8 | 39.7 | 9.3 | 3.1 | 2.1 | 0.7 | 5.4 | 1.2 | 23.0 |
| • DeYO | 28.1 | 3.7 | 7.2 | 0.9 | 0.1 | 0.1 | 0.1 | 0.1 | 0.1 | 0.1 | 0.1 | 0.1 | 0.1 | 0.1 | 0.1 | 2.8 |
| • TTE (w. DeYO) | **43.0** | **31.7** | **52.7** | **52.3** | **45.1** | **36.4** | **52.2** | **53.6** | **40.9** | **43.0** | **59.1** | **60.3** | **47.4** | **68.9** | **59.0** | **49.7** |
| ViTBase | 9.4 | 29.1 | 15.8 | 43.9 | 6.7 | 23.4 | 26.3 | 30.5 | 8.3 | 34.0 | 47.4 | 44.5 | 27.0 | 54.7 | 47.6 | 29.9 |
| • Tent | 30.9 | 18.5 | 7.1 | 0.1 | 0.1 | 0.1 | 0.1 | 0.1 | 0.1 | 0.1 | 0.1 | 0.1 | 0.1 | 0.2 | 0.1 | 3.9 |
| • CoTTA | 34.1 | 13.6 | 2.2 | 0.1 | 0.1 | 0.1 | 0.1 | 0.1 | 0.1 | 0.1 | 0.1 | 0.1 | 0.1 | 0.1 | 0.1 | 3.4 |
| • SAR | 46.6 | 54.2 | 54.7 | 49.0 | 38.7 | 38.9 | 45.3 | 45.3 | 37.5 | 40.0 | 45.8 | 50.0 | 37.3 | 58.6 | 48.6 | 46.0 |
| • DeYO | 49.1 | 36.7 | 63.3 | 61.4 | 52.8 | 49.9 | 59.1 | 62.2 | 48.9 | 54.0 | 61.5 | 69.2 | 2.2 | 64.8 | 67.2 | 53.5 |
| • TTE (w. DeYO) | **54.0** | **55.2** | **64.6** | **62.5** | **54.2** | **55.9** | **62.6** | **66.8** | **52.0** | **59.1** | **68.5** | **69.9** | **55.2** | **74.9** | **68.4** | **61.6** |

Table 4: TTA with natural distribution shifts. Classification accuracy (%) with ImageNet-Sketch (S), ImageNet-Rendition (R), ImageNet-V2 (V2). **Bold** numbers are the best results.

| (a) Label Shifts | | | | | (b) Batch Size 1 | | | |
|---|---|---|---|---|---|---|---|---|
| Methods | S | R | V2 | Avg | Methods | S | R | V2 | Avg |
|---|---|---|---|---|---|---|---|---|---|
| ResNet50-GN | 29.2 | 40.8 | 68.9 | 46.3 | ResNet50-GN | 29.2 | 40.8 | 68.9 | 46.3 |
| • Tent | 30.8 | 41.5 | **68.9** | 47.1 | • Tent | 31.9 | 42.1 | 68.9 | 47.6 |
| • CoTTA | 30.8 | 41.2 | 68.8 | 46.9 | • CoTTA | 26.7 | 40.6 | 68.9 | 45.3 |
| • SAR | 30.6 | 41.6 | **68.9** | 47.0 | • SAR | 31.6 | 41.9 | 68.9 | 47.5 |
| • DeYO | 34.6 | 44.4 | **68.9** | 49.3 | • DeYO | 37.0 | 46.0 | **69.0** | 50.7 |
| • TTE (w. DeYO) | **36.9** | **45.1** | **68.9** | **50.3** | • TTE (w. DeYO) | **39.1** | **47.1** | **69.0** | **51.7** |
| ViTBase | 18.2 | 43.1 | 66.2 | 37.9 | ViTBase | 18.2 | 43.1 | 66.2 | 37.9 |
| • Tent | 8.8 | 41.9 | 68.9 | 39.8 | • Tent | 7.0 | 40.6 | 69.1 | 38.9 |
| • CoTTA | 28.4 | 45.1 | 67.5 | 47.0 | • CoTTA | 24.0 | 46.0 | 66.9 | 45.6 |
| • SAR | 17.8 | 45.1 | 68.5 | 43.8 | • SAR | 26.4 | 44.7 | 69.4 | 46.8 |
| • DeYO | 42.4 | 58.6 | **71.1** | 57.3 | • DeYO | 43.6 | 59.8 | 71.2 | 58.2 |
| • TTE (w. DeYO) | **43.6** | **59.1** | **71.1** | **57.9** | • TTE (w. DeYO) | **45.0** | **61.3** | **71.3** | **59.2** |

even further when paired models have stronger representations. TTE would have potential to boost performance in future TTA methods.

**Continual TTA with non-i.i.d. conditions**. We extended the ImageNet-C scenarios to continual TTA with non-i.i.d. distributions and classes. These challenging scenarios require models to adapt to imbalanced classes (Label Shifts) across 15 corruptions continuously, a setting where TTA models are prone to collapse. For comparison, we included CoTTA (Wang et al., 2022), originally designed for continual TTA with a teacher-student framework. Table 3 reports classification accuracy, with detailed results and standard deviations in Table 16. TTE consistently achieved the highest average accuracy, with 49.7% on ResNet50-GN and 61.6% on ViTBase, while other methods, including DeYO, showed unstable and near-zero accuracy in later adaptation stages (all comparative methods on ResNet50-GN and Tent, CoTTA on ViTBase). The results highlight that ensemble strategies improve model representations for better adaptation, while robust knowledge distillation stabilizes TTA optimization and prevents model collapse. Furthermore, we conducted these experiments with other baseline methods integrated with TTE, as shown in Table 11, and compared their performance in addressing the issue of catastrophic forgetting in Figure 8.

**Natural distribution shifts**. We evaluated adaptation under natural distribution shifts by combining two challenging scenarios: Label Shifts and Batch Size 1. Table 4 presents classification accuracy for both ResNet50-GN and ViTBase. TTE proved highly effective on natural distribution shifts, consistently outperformed DeYO with a +1.0% gain on ResNet50-GN and +0.6% on ViTBase for Label Shifts, and a +1.0% on ResNet50-GN and +1.0% on ViTBase in the Batch Size 1. Notably, the ensemble strategies demonstrated strong benefits on ImageNet-R where natural shifts occur concurrently, outperforming DeYO across all cases. However, in ImageNet-V2, which features intrinsic shifts, TTE provided modest gains (0.0% to 0.1%) compared to DeYO, suggesting that further investigation is needed for this type of shift. Additionally, Table 12 reports the results of other baseline methods integrated with TTE.

**Computational complexity**. Integrating TTE required only an additional feedforward pass for $f_e$, resulting in minimal computational overhead. Table 9 in Appendix compares the computational complexity with baseline models. Further discussion is provided in Appendix C.3.

Table 5: Impacts of TTE components on classification accuracy (%) in Label Shifts with ImageNet-C. DB: de-biasing in knowledge distillation, WSE: weight-space ensemble with adaptive momentum, RKL: reverse KL divergence, Do: dropout ensemble.

| Methods | Gauss | Shot | Impul | Defoc | Glass | Motio | Zoom | Snow | Frost | Fog | Brit | Contr | Elast | Pixel | JPEG | Avg |
|---|---|---|---|---|---|---|---|---|---|---|---|---|---|---|---|---|
| ResNet50-GN | 18.0 | 19.8 | 17.9 | 19.8 | 11.4 | 21.4 | 24.9 | 40.4 | 47.3 | 33.6 | 69.3 | 36.3 | 18.6 | 28.4 | 52.3 | 30.6 |
| DeYO | 28.1 | 44.3 | 42.9 | 23.4 | 16.6 | 41.5 | 6.1 | 52.9 | 52.0 | 20.2 | 73.2 | 53.0 | 37.7 | 60.0 | 59.4 | 40.8 |
| +DB | 32.8 | 36.3 | 35.9 | 23.7 | 20.8 | 35.6 | 32.9 | 49.9 | 47.1 | 51.7 | 72.6 | 49.0 | 36.6 | 55.2 | 57.6 | 42.5 |
| +DB+WSE | 42.5 | 44.7 | 43.6 | 34.3 | 32.7 | 42.8 | 44.2 | 53.8 | 52.5 | 59.2 | 73.2 | 53.5 | 49.4 | 59.5 | 59.7 | 49.7 |
| +DB+WSE+RKL | 42.6 | 44.9 | 43.9 | **34.4** | 33.0 | 42.9 | 44.4 | 54.0 | 52.7 | 59.5 | 73.3 | 53.7 | 49.9 | 59.6 | 59.7 | 49.9 |
| +DB+WSE+RKL+DO | **43.0** | **45.3** | **44.0** | **34.4** | **33.6** | **43.4** | **46.0** | **55.2** | **53.6** | **61.0** | **73.4** | **54.4** | **51.5** | **61.0** | **60.2** | **50.7** |
| ViTBase | 9.4 | 6.7 | 8.3 | 29.1 | 23.4 | 34.0 | 27.0 | 15.8 | 26.3 | 47.4 | 54.7 | 43.9 | 30.5 | 44.5 | 47.6 | 29.9 |
| DeYO | 49.1 | 35.9 | 53.6 | 57.6 | 58.6 | 63.8 | 37.5 | 67.9 | 66.0 | 73.1 | 77.9 | 66.5 | 68.6 | 73.5 | 70.1 | 61.3 |
| +DB | 49.3 | 47.8 | 50.4 | 57.0 | 56.1 | 60.7 | 55.8 | 64.9 | 64.3 | 71.7 | 78.0 | 66.5 | 65.0 | 72.0 | 68.9 | 61.9 |
| +DB+WSE | 53.0 | 50.7 | 54.1 | 58.7 | 59.0 | 63.3 | 59.6 | 67.3 | 66.1 | 73.4 | 78.2 | **68.5** | 68.2 | 73.2 | 69.9 | 64.2 |
| +DB+WSE+RKL | 53.4 | 53.3 | 54.4 | 58.9 | 59.3 | 63.8 | 59.6 | 67.7 | 66.4 | 73.7 | **78.2** | **68.5** | 68.5 | 73.4 | 70.1 | 64.6 |
| +DB+WSE+RKL+DO | **54.0** | **54.7** | **55.2** | **58.9** | **59.7** | **64.5** | **62.1** | **68.3** | **66.7** | **73.9** | 78.0 | 68.3 | **69.3** | **73.8** | **70.3** | **65.2** |

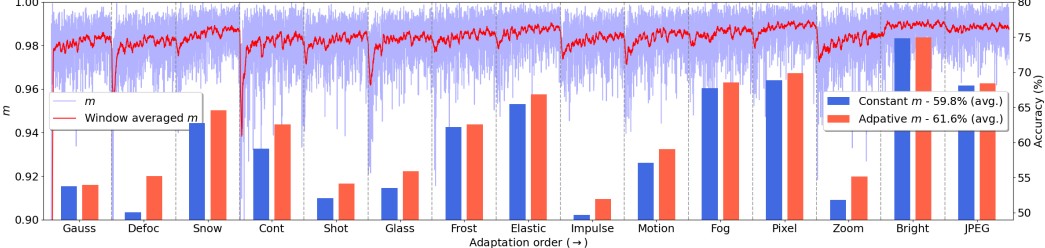

Figure 5: Momentum profile in Continual TTA with non i.i.d. conditions (left axis). Classification accuracy (%) of TTE, reported to compare constant $m$ and adaptive $m$ in this setup. (right axis)

## 4.2 ABLATION STUDY

**Impact of the proposed components**. We analyzed the contribution of each proposed component to improving the baseline DeYO. Table 5 shows classification accuracy as components are added sequentially. First, we constructed a conventional teacher-student framework with DeYO, updated using EMA with $m = 0.999$ and standard KL divergence for knowledge distillation, which are the standard setting in such frameworks (Performance for this setup is not included). Adding de-biasing in knowledge distillation (+DB) prevented collapse and led to general gains of +1.7% on ResNet50-GN and +0.6% on ViTBase in average. Introducing the weight-space ensemble with adaptive momentum (+WSE) improved performance across all distributions, with average gains of +7.2% on ResNet50-GN and +2.3% on ViTBase. Replacing the standard KL divergence with reverse KL (+RKL) provided the average gains of +0.2% on ResNet50-GN and +0.4% on ViTBase. Lastly, applying dropout to $f_a$ (+DO) further improved average accuracy by +0.8% on ResNet50-GN and +0.6% on ViTBase.

**Adaptive momentum**. Table 6 compares the proposed adaptive momentum scheme with the conventional constant scheme across a wide range of $m$ values in Label Shifts for TTE with DeYO. The commonly used value of $m = 0.999$ from other momentum-based approaches (Wang et al., 2022; Yuan et al., 2023) did not yield significant ensemble effects, resulting in lower accuracy. In contrast, lower momentum values enhanced the ensemble effect, leading to better performance—specifically $m = 0.5$ for ResNet50-GN and $m = 0.99$ for ViTBase. The proposed adaptive momentum scheme, which lower momentum based on probabilistic distance, achieved the best performance for ViTBase. For ResNet, lower momentum generally worked well. These results are consistent with our preliminary study, which observed *linear mode connectivity* between TTA models.

Table 6: Classification accuracy (%) with varying momentum values in TTE (ImageNet-C, level 5).

| | ResNet50-GN | ViTBase |
|---|---|---|
| DeYO | 40.8 | 61.3 |
| $m = 0.999$ | 43.0 | 62.0 |
| $m = 0.99$ | 49.3 | 65.0 |
| $m = 0.9$ | 50.6 | 64.8 |
| $m = 0.5$ | **50.7** | 64.6 |
| Adaptive $m$ | 50.7 | **65.2** |

To further verify effectiveness of the adaptive scheme, we tested it in the challenging scenario of Continual TTA with non-i.i.d. conditions for ViTBase, where distribution shifts were sequentially changed. Figure 5 illustrates the momentum values during TTA and compares classification accu-

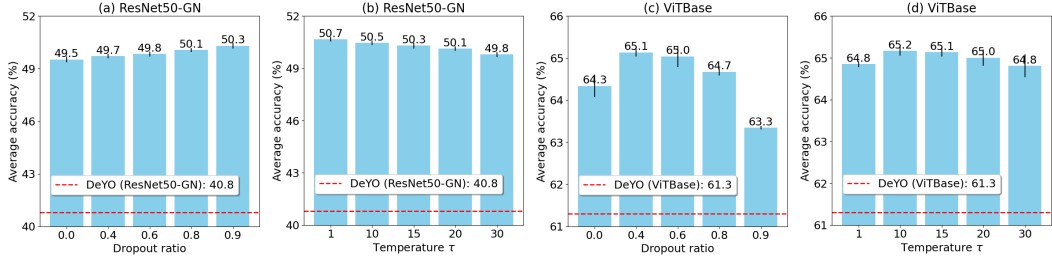

Figure 6: Hyperparameter Sensitivity. Average classification accuracy (%) in Label Shifts (ImageNet-C, level 5). If not specified in experiments, the default values were: dropout ratio = 0.9, $\tau = 15.0$ for ResNet50-GN and dropout ratio = 0.4, $\tau = 15.0$ for ViTBase.

racy between constant and adaptive schemes. Note that the constant momentum value was set to $m = 0.99$, as it achieved the best performance in Table 6. As shown in Figure 5, momentum values were adjusted to lower levels during transitions to new distributions, actively promoting ensemble construction. This dynamic adjustment led to a +1.8% accuracy improvement. For additional study, the proposed adaptive weight averaging is compared with stochastic weight averaging (Izmailov et al., 2018) in Appendix D.5.

**Ensemble hyperparameter sensitivity**. Figure 6 illustrates the impact of varying ensemble hyperparameters (with DeYO) on ViTBase and ResNet. It is noteworthy that TTE steadily outperformed DeYO across different hyperparameter values, demonstrating the low sensitivity of hyperparameter. For dropout, increasing the ratio generally enhanced accuracy compared to a ratio of 0.0. However, the extreme case of a high dropout ratio above 0.9 led to decreased accuracy in ViTBase while led to increased accuracy in ResNet. For the temperature $\tau$ controlling decaying factor in adaptive momentum, TTE achieved the best performance with $\tau = 1.0$ for ResNet and $\tau = 10.0$ for ViTBase.

**Analysis for de-biasing scheme**. The de-biasing scheme, which penalizes biased representations in knowledge distillation, is more closely related to optimization stability than to adaptation capability. To assess optimization stability, Figure 7 presents the standard deviation of performance across three random seeds. As $\alpha$, controlling de-biasing degree in $\hat{y}'_e$, approached 1.0, classification accuracy dropped and standard deviation increased, indicating the models struggled with biasing under certain random seeds. When the bias update rate $n$ approached 1.0, accuracy decreased with greater standard deviation, suggesting the update was too slow to estimate bias properly. Extensive experiments with $\alpha = 3.0$ and $n = 0.99$ con-

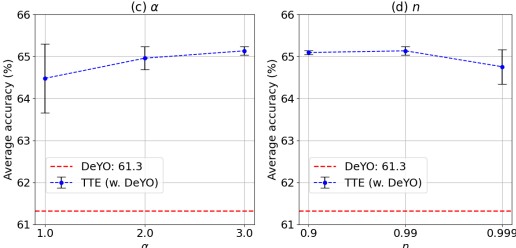

Figure 7: Average classification accuracy (%) with standard deviation in Label Shifts for ViT-Base, with varying hyperparameters in de-biasing scheme. The values for ensemble hyperparameters were set to dropout ratio=0.4 and $\tau = 15.0$.

firmed the de-biasing scheme effectively prevented model collapse in diverse test-time scenarios. However, we observed the sensitivity of the proposed de-biasing scheme with true-biased scenario where same class samples were fed into the model consecutively over 100 times. Further analysis is included in Appendix D.6.

## 5 CONCLUSION

We proposed a novel test-time adaptation (TTA) approach, TTE, that enhances the performance of existing methods through ensemble strategies. TTE consistently improved adaptation, demonstrated broad applicability, and remained computationally efficient. Extensive experiments confirmed its effectiveness, and notably, TTE exhibited exceptional stability, preventing model collapse across four datasets and four test-time scenarios. This approach holds significant potential for advancing future TTA developments.

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

# Appendix

# Contents

## A  RELATED WORK

**Test-time adaptation** (TTA), using pretrained models, has become essential for adjusting to distribution shifts in test data (Wang et al., 2021). Many existing methods have focused on optimizing stability to prevent model collapse amid dynamic distribution shifts, by selecting reliable samples (Niu et al., 2023; 2022; Yuan et al., 2023), ensuring consistency in predictions (Wang et al., 2022; Zhang et al., 2022b; Chen et al., 2022), manipulating affine transform statistics (Gong et al., 2022; Lim et al., 2023; Zhao et al., 2023) and using a robust optimizer (Niu et al., 2023; Gong et al., 2024). However, enhancing model representation for adaptation capability remains less explored to date. Similar to the proposed TTE, several researchers have adopted student-teacher networks and employed an exponential moving average strategy to update the teacher network (Wang et al., 2022; Yuan et al., 2023; Döbler et al., 2023; Chen et al., 2022). Unlike TTE, however, they typically have introduced a weight-averaged network to ensure optimization stability and minimize abrupt model changes by using a high momentum value for the average, close to 0.999. To achieve relevant representations, these methods have often utilized dense image augmentation or relied on source data. Unfortunately, such strategies resulted in increased computational complexity.

**Test-time training**, similar to TTA, has been designed to actively adjust a pretrained model to distribution shifts using test data. Unlike TTA, which domain adaptation processes are independent of the training phase, test-time training starts by developing a training process that can be extended into a test phase for adaptation (Sun et al., 2020). Most strategies have implemented a combined optimization of supervised and self-supervised learning during a training phase, subsequently performing the self-supervised learning with test data (Sun et al., 2020; Liu et al., 2021; Gandelsman et al., 2022). However, dependence on specific training procedures and architectures, designed for self-supervised objectives, could restrict the methods' applicability to various off-the-shelf models. Additionally, these methods sometimes demanded substantial computational resources due to the intensive augmentations used in self-supervised learning (Gandelsman et al., 2022).

**Fine-tuning under linear approximation** has garnered significant attention as an efficient strategy for adapting complex pretrained models to downstream tasks. Research has shown that the later stages of training deep neural networks often stabilize within nearly-convex regions, suggesting that the landscape of the models' cost functions is more tractable (Izmailov et al., 2018; Frankle et al., 2020; Fort et al., 2020). This observation persists even when fine-tuning large networks with datasets much smaller than those required to train a model from scratch. Further advancing this concept, fine-tuning processes have been approximated by a first-order Taylor expansion, transforming the process into a linear system based on Neural Tangent Kernel (NTK) features (Maddox et al., 2021; Jacot et al., 2018). More recently, notable approaches have proposed averaging the weights of various fine-tuned models, effectively replicating the performance of output ensembles across both in-distribution and out-of-distribution datasets, and underscoring the near-linear characteristics of the fine-tuning process (Wortsman et al., 2022; Rame et al., 2022). Inspired by these intriguing findings, we aim to develop a TTA method based on the premise that test-time tuning of pretrained models can also exhibit near-linear characteristics.

**Constructing versatile representations** is crucial for improving generalization performance, especially in distribution shifts. Researchers have increased the versatility of representations by diversifying architectures, datasets, and hyper-parameters (Dvornik et al., 2020; Chowdhury et al., 2021; Ganaie et al., 2022), rather than merely enlarging dataset sizes. Some studies have collected features from different models (Li et al., 2023), while others have integrated diverse representations through weight averaging (Wortsman et al., 2022; Rame et al., 2022). Notably, recent research has shown that redundant representations, although not beneficial for in-distribution performance, can significantly enhance out-of-distribution generalization (Zhang & Bottou, 2023). Building on this concept, the proposed components in TTE aim to cultivate versatile representations during test time, thereby enhancing performance against unexpected distribution shifts.

# B  DETAILS OF THE PRELIMINARY STUDY

## B.1  THEORETICAL ANALYSIS OF LINEAR MODE CONNECTIVITY IN TTA

Our preliminary experiments suggest that weight-space interpolation can mimic the effects observed in output-space interpolation. It is crucial, however, to understand the conditions under which weight-space interpolation can theoretically approximate output-space interpolation. Previous research (Mu et al., 2020; Maddox et al., 2021) has proposed approximating the fine-tuning process using a first-order Taylor expansion, resulting in a linear system that operates on Neural Tangent Kernel (NTK) features (Jacot et al., 2018; Fort et al., 2020). We extend this hypothesis to TTA, a specific form of fine-tuning, positing that it can similarly be approximated by a linear expansion. Under this framework, weight-space interpolation approximates output-space interpolation correctly.

**Proposition**. If for any $\theta$ within $\Theta = \{(1 - \alpha)\theta_1 + \alpha\theta_2 : \alpha \in [0, 1]\}$, the function $f(\theta)$ can be approximated linearly around $\theta_0$ as follows: $f(\theta) = f(\theta_0) + \nabla f(\theta_0)^\top (\theta - \theta_0)$, then weight-space interpolation between $\theta_1$ and $\theta_2$ is equivalent to performing an output-space interpolation. Here, $\theta_0$ denotes the parameter of a pretrained model.

*Proof.* We initiate with output-space ensemble and retrieve weight-space ensemble

$$
\begin{aligned}
&(1 - \alpha)f(\theta_1) + \alpha f(\theta_2) \\
&= (1 - \alpha)f(\theta_0) + (1 - \alpha)\nabla f(\theta_0)^\top (\theta_1 - \theta_0) + \alpha f(\theta_0) + \alpha \nabla f(\theta_0)^\top (\theta_2 - \theta_0) \\
&= f(\theta_0) + \nabla f(\theta_0)^\top ((1 - \alpha)(\theta_1 - \theta_0) + \alpha(\theta_2 - \theta_0)) \\
&= f(\theta_0) + \nabla f(\theta_0)^\top ((1 - \alpha)\theta_1 + \alpha\theta_2 - \theta_0)) \\
&= f((1 - \alpha)\theta_1 + \alpha\theta_2)
\end{aligned}
\tag{6}
$$

$\square$

## B.2  PRELIMINARY STUDY FOR LINEAR MODE CONNECTIVITY

We conducted a preliminary study to empirically investigate the linear connectivity between two TTA models and to suggest strong cues for the benefits of weight-space interpolation for TTA. All experiments in the preliminary study were performed with ImageNet-C (severity level 5) by using Tent (Wang et al., 2021). The sample selection scheme suggested in Niu et al. (2023) was integrated with Tent to prevent model collapse and to clearly analyze the linear property in TTA. The details of experimental procedures are as follows:

1. We performed TTA for four corruptions individually, which were Gaussian noise, defocus blur, snow, and contrast.

2. Two TTA models were interpolated with a mixing coefficient $\alpha$ in both output and weight spaces.

3. Classification accuracy was measured over all 15 corruptions in ImageNet-C (level 5), by using the predictions from the two types of interpolation.

4. We repeated the second and the third steps by changing $\alpha$ in $[0, 1]$.

Accuracy values averaged across two target corruptions were used as $x$ values in Figure 1, to show adaptation performance. Accuracy over all other corruptions were used as $y$ values to show generalization performance.

## B.3  EXPANDING TO VARIOUS TTA SCENARIOS

Inspired by the results in Appendix B.2, we conducted an additional preliminary study to verify the effects of linear mode connectivity across various TTA scenarios. In this study, four corruptions (Gaussian noise, defocus blur, snow, and contrast) were selected as target distributions, while the remaining corruptions were considered as non-target distributions. We assessed weight-space interpolation through two scenarios: **single-instance TTA**, where each model adapts to one corruption,

and **continual TTA**, where a model sequentially updates across four target distributions, reflecting practical settings. The classification accuracy for both target and non-target corruptions was measured after TTA processes.

**Single-instance TTA**. We constructed a ensemble by averaging the parameters of four models with 0.25 mixing coefficients. Table 7 compares the performance of each TTA model and their ensemble. Interestingly, the ensemble model achieved the best average performance on both target and non-target corruptions, suggesting that merely averaging model parameters can effectively capture diverse representations from multiple models.

**Continual TTA**. Model parameters were saved before transitioning to subsequent corruptions, yielding three intermediate models plus a final model. Table 8 compares the performance of the final model against the ensemble of the four models with 0.25 coefficients. The ensemble models generally outperformed the continual models, demonstrating that linear mode connectivity remains valid in continual setups. Notably, even with the sample selection scheme in the baseline TTA models, the continual model collapsed in the sequence of D→ S→ C→ G, providing near-zero accuracy, while the ensemble model mitigated the performance degradation.

Table 7: Classification accuracy (%) under single-instance TTA. Note that WSE stands for weight-space ensemble. G,D,S and C represent Gaussian noise, defocus blur, snow and contrast respectively. Bold and underlined numbers are the best and the second best results.

| Method | Target corruptions | | | | | Non-target corruptions | | | | | | | | | | | |
| --- | --- | --- | --- | --- | --- | --- | --- | --- | --- | --- | --- | --- | --- | --- | --- | --- | --- |
| | Gauss | Defoc | Snow | Contr | Avg | Shot | Impul | Glass | Motion | Zoom | Frost | Fog | Brit | Elastic | Pixel | JPEG | Avg |
| No adapt | 9.4 | 29.1 | 15.8 | 43.9 | 24.6 | 6.7 | 8.3 | 23.4 | 34.0 | 27.0 | 26.3 | 47.4 | 54.7 | 30.5 | 44.5 | 47.6 | 31.8 |
| TTA (G) | **51.0** | 28.4 | 42.3 | 29.1 | 37.7 | **47.8** | **51.7** | 29.9 | 38.8 | 32.4 | 49.0 | 44.2 | 70.2 | 39.5 | 57.1 | 57.1 | 47.1 |
| TTT (D) | 12.9 | **57.2** | 38.5 | 52.5 | 40.3 | 11.8 | 12.1 | **35.2** | **53.4** | **40.4** | 42.1 | 60.0 | 71.7 | 34.4 | 57.3 | 57.4 | 43.2 |
| TTA (S) | 21.0 | 35.5 | **60.9** | 37.4 | 38.7 | 16.3 | 20.8 | 31.4 | 43.3 | 32.0 | **58.2** | 52.6 | **74.7** | **40.8** | 55.7 | 57.8 | 44.0 |
| TTA (C) | 14.0 | 43.5 | 29.0 | **67.7** | 38.6 | 11.4 | 13.1 | 30.2 | 46.1 | 34.7 | 39.2 | **61.4** | 65.5 | 32.2 | 53.9 | 56.5 | 40.4 |
| WSE (G+D+S+C) | 30.7 | 45.7 | 49.5 | 56.0 | **45.5** | 26.2 | 31.6 | 34.5 | 49.5 | 38.2 | 51.1 | 59.3 | 73.7 | 38.7 | **60.6** | **60.5** | **47.6** |

Table 8: Classification accuracy (%) under continual TTA with four corruption orders. Notations are identical to Table 7.

| Method | Target corruptions | | | | | Non-target corruptions | | | | | | | | | | | |
| --- | --- | --- | --- | --- | --- | --- | --- | --- | --- | --- | --- | --- | --- | --- | --- | --- | --- |
| | Gauss | Defoc | Snow | Contr | Avg | Shot | Impul | Glass | Motion | Zoom | Frost | Fog | Brit | Elastic | Pixel | JPEG | Avg |
| No adapt | 9.4 | 29.1 | 15.8 | 43.9 | 24.6 | 6.7 | 8.3 | 23.4 | 34.0 | 27.0 | 26.3 | 47.4 | 54.7 | 30.5 | 44.5 | 47.6 | 31.8 |
| TTA (G→ D→ S→ C) | 38.7 | **52.5** | 40.2 | **66.1** | 49.4 | 36.9 | 39.9 | 35.0 | **50.5** | 33.1 | 51.0 | **60.1** | 75.2 | 38.3 | 62.4 | **63.5** | 49.6 |
| WSE (G+D+S+C) | **45.9** | 51.7 | **49.1** | 56.1 | **50.7** | **43.5** | **46.8** | **35.9** | 50.4 | **36.4** | **53.8** | 58.6 | **75.4** | **39.9** | **63.3** | 62.9 | **51.5** |
| TTA (C→ G→ D→ S) | 40.4 | 46.4 | 42.5 | 51.9 | 45.3 | 38.1 | 40.8 | 27.4 | 44.9 | 27.9 | 51.7 | 52.1 | 74.2 | 35.2 | 60.1 | 62.4 | 46.8 |
| WSE (C+G+D+S) | **43.6** | **51.3** | **49.9** | **64.1** | **52.2** | **39.8** | **44.3** | **35.3** | **51.4** | **38.3** | **52.5** | **61.8** | **74.5** | **38.5** | **62.5** | **63.3** | **51.1** |
| TTA (S→ C→ G→ D) | **44.0** | **55.8** | 56.3 | 62.2 | **54.6** | **41.2** | **44.5** | **38.1** | **53.7** | **41.4** | 52.8 | **61.3** | **75.8** | 41.5 | 61.4 | 63.7 | **52.3** |
| WSE (S+C+G+D) | 40.0 | 48.6 | **60.0** | **64.0** | 53.1 | 35.5 | 40.7 | 37.8 | 51.6 | 39.4 | **56.9** | 60.9 | 75.6 | **42.5** | **62.7** | **63.8** | 51.6 |
| TTA (D→ S→ C→ G) | 0.2 | 0.3 | 0.2 | 0.1 | 0.2 | 0.2 | 0.1 | 0.2 | 0.1 | 0.1 | 0.4 | 0.1 | 9.9 | 0.6 | 2.3 | 3.7 | 1.6 |
| WSE (D+S+C+G) | **2.4** | **16.9** | **3.5** | **6.8** | **7.4** | **2.8** | **1.9** | **4.9** | **11.5** | **4.9** | **18.5** | **13.1** | **62.3** | **11.6** | **44.8** | **44.1** | **20.0** |

## C  DETAILS OF IMPLEMENTATION

### C.1  BASELINE MODELS

We utilized pre-trained ViT-base[1] and ResNet50-GN[2] obtained from the publicly available PyTorch Image Models repository Wightman (2019). The public models used were trained on ImageNet-1k for image recognition tasks. The implementations of the comparative methods were obtained from their public repositories and followed the guidelines outlined in their original papers. Additionally, when integrating TTE with existing TTA approaches, we followed the original implementations and hyperparameter settings as specified in the respective papers to ensure accurate evaluation of TTE's effects. Further details are provided below.

**Tent** (Wang et al., 2021)[3] **Tent** used stochastic gradient descent (SGD) with a momentum of 0.9 as the optimizer. The learning rate was set to 0.00025 for ResNet50-GN and 0.001 for ViTBase. For a batch size of 1, the learning rates were adjusted to 0.00025/32 for ResNet50-GN and 0.001/64 for ViTBase. The trainable parameters included all affine parameters in the normalization layers.

**CoTTA** (Wang et al., 2022)[4] used the Adam optimizer with a learning rate of 0.0025 for ResNet and 0.001 for ViTBase. The method employed model-specific hyperparameters and reported their values for ResNet, prompting us to search for optimal values for ViTBase. The restoration factor $p$ was explored within the range [0.01,0.9], and the EMA smoothing factor $\alpha$ was searched within [0.1,0.001]. Based on performance, we selected $p = 0.7$ and $\alpha = 0.001$ for ViTBase, while for ResNet, we followed the author's recommendation with $p = 0.01$ and $\alpha = 0.001$. Additionally, the augmentation confidence threshold $p_{th}$ was set to 0.1. All weights in the architectures were trainable.

**SAR** (Niu et al., 2023)[5] used SGD with a momentum of 0.9 as the optimizer. The learning rate was set to 0.00025 for ResNet50-GN and 0.001 for ViTBase. For a batch size of 1, the learning rates were adjusted to 0.00025/16 for ResNet50-GN and 0.001/32 for ViTBase. We set the sharpness threshold to $\rho = 0.05$ and the entropy threshold to $E_0 = 0.4 \cdot \ln(1000)$. The learnable parameters included affine parameters in the normalization layers, while the top layers were frozen: layer 4 in ResNet and blocks 9-11 in ViTBase.

**DeYO** (Lee et al., 2024)[6] used SGD with a momentum of 0.9 as the optimizer. The learning rate was set to 0.00025 for ResNet50-GN and 0.001 for ViTBase. For a batch size of 1, the learning rates were adjusted to 0.00025/16 for ResNet50-GN and 0.001/32 for ViTBase. The required hyperparameters for DeYO are the entropy threshold $\tau_{Ent}$, the probability difference threshold $\tau_{PLPD}$, and the normalizing factor $Ent_0$. We set $\tau_{Ent} = 0.5 \times \ln(1000)$, $\tau_{PLPD} = 0.3$, and $Ent_0 = 0.4 \times \ln(1000)$. The learnable parameters included the affine parameters in the normalization layers, while the top layers were frozen: layer 4 in ResNet and blocks 9-11 in ViTBase.

**+ TTE (Ours)** employed SGD with a momentum of 0.9 as the optimizer. The learning rate was set to 0.00025 for ResNet50-GN and 0.001 for ViTBase. For a batch size of 1, the learning rates were adjusted to $0.00025/16$ for ResNet50-GN and $0.001/32$ for ViTBase. The temperature $\tau$ was set to 1.0 with a dropout ratio of 0.9 for ResNet, and $\tau = 10.0$ with a dropout ratio of 0.4 for ViTBase. For the de-biasing scheme, $\alpha = 3.0$ and $n = 0.99$ were used. Note that a linear ramp-up was applied to gradually increase $m$ from 0.0 to $m_0$ over initial 100 iterations for a batch size of 64, and initial 6400 iterations for a batch size of 1, where ensemble effects were insignificant. After the ramp-up phase, the active momentum scheme was applied. The trainable parameters in TTE were identical to those of the baseline model with which TTE was integrated.

---

[1]https://storage.googleapis.com/vit_models/augreg/B_16-i21k-300ep-lr_0.001-aug_medium1-wd_0.1-do_0.0-sd_0.0--imagenet2012-steps_20k-lr_0.01-res_224.npz

[2]https://github.com/rwightman/pytorch-image-models/releases/download/v0.1-rsb-weights/resnet50_gn_a1h2-8fe6c4d0.pth

[3]https://github.com/DequanWang/tent

[4]https://github.com/qinenergy/cotta

[5]https://github.com/mr-eggplant/SAR

[6]https://github.com/Jhyun17/DeYO

## C.2 ALGORITHMS

TTE was integrated into three existing methods to verify its effectiveness and broad applicability: Tent (Wang et al., 2021), SAR (Niu et al., 2023), and DeYO (Lee et al., 2024). TTE was seamlessly incorporated by leveraging the optimization procedures and sample selection/weighting mechanisms originally proposed in these methods. The algorithms for Tent, SAR, and DeYO are detailed in Algorithm 1, 2, and 3, respectively, with the parts introduced by TTE highlighted in blue. Importantly, TTE required few implementation lines, adding minimal computational burden.

---

**Algorithm 1:** TTE + Tent (Wang et al., 2021)

---

**Input:** Test samples $\mathcal{D}_{\text{test}} = \{\boldsymbol{x}_i\}_{i=1}^M$, adapter model $f_a(\cdot; \theta_a, \rho)$ with trainable parameters $\tilde{\theta}_a \subset \theta_a$ and a dropout ratio $\rho$, ensemble model $f_e(\cdot; \theta_e)$ with $\tilde{\theta}_e$ aligned to $\tilde{\theta}_a$, bias vector $\boldsymbol{c}_{bias}$, step size $\eta > 0$, momentum $m$, debiasing parameters $n, \alpha$.

**Output:** Predictions $\{\hat{\boldsymbol{y}}_{e,i}\}_{i=1}^M$.

1 Initialize $\tilde{\theta}_a \leftarrow \tilde{\theta}_0, \tilde{\theta}_e \leftarrow \tilde{\theta}_0, c_{bias} \leftarrow \mathbf{0}$;

2 **for** $\boldsymbol{x}_i \in \mathcal{D}_{test}$ **do**

3      Predict $\hat{\boldsymbol{y}}_{a,i} = f_a(\boldsymbol{x}_i; \theta_a, \rho), \hat{\boldsymbol{y}}_{e,i} = f_e(\boldsymbol{x}_i; \theta_e)$ and compute entropy $E(\hat{\boldsymbol{y}}_{a,i}; \theta_a)$;

4      Compute weight $w(\hat{\boldsymbol{y}}_{e,i})$ and de-biased representation $\hat{\boldsymbol{y}}'_{e,i}$;

5      Compute $KL(\hat{\boldsymbol{y}}_{a,i} || \hat{\boldsymbol{y}}'_{e,i}; \theta_a)$ and total objective $\mathcal{L}_{tte}$:

$$\mathcal{L}_{tte}(\hat{\boldsymbol{y}}_{a,i}, \hat{\boldsymbol{y}}'_{e,i}; \theta_a) = E(\hat{\boldsymbol{y}}_{a,i}; \theta_a) + KL(\hat{\boldsymbol{y}}_{a,i} || \hat{\boldsymbol{y}}'_{e,i}; \theta_a)$$

6      Compute gradient $g = \nabla_{\tilde{\theta}_a} \mathcal{L}_{tte}(\hat{\boldsymbol{y}}_{a,i}, \hat{\boldsymbol{y}}'_{e,i}; \theta_a)$;

7      Update $\tilde{\theta}_a \leftarrow \tilde{\theta}_a - \eta g$;

8      Compute momentum $m$ from Eq.2 and update ensemble parameters: $\tilde{\theta}_e \leftarrow m \cdot \tilde{\theta}_e + (1 - m) \cdot \tilde{\theta}_a$;

9      Compute first-order batch statistics $\mu_{\hat{\boldsymbol{y}}_{a,i}} = \frac{1}{N} \sum_{i=1}^N \hat{\boldsymbol{y}}_{a,i}$;

10      Update bias vector: $c_{bias} \leftarrow n \cdot c_{bias} + (1 - n) \cdot \mu_{\hat{\boldsymbol{y}}_{a,i}}$;

11 **end**

---

**Algorithm 2:** TTE + SAR (Niu et al., 2023)

---

**Input:** Test samples $\mathcal{D}_{\text{test}} = \{\boldsymbol{x}_i\}_{i=1}^M$, model $f_a(\cdot; \theta_a, \rho)$ with trainable parameters $\tilde{\theta}_a \subset \theta_a$ and a dropout ratio $\rho$, ensemble model $f_e(\cdot; \theta_e)$ with $\tilde{\theta}_e$ aligned to $\tilde{\theta}_a$, bias vector $\boldsymbol{c}_{bias}$, step size $\eta > 0$, neighborhood size $\rho > 0$, $\tau_{\text{Ent}} > 0$ in Eq. (2), $e_0 > 0$ for model recovery. momentum $m$, debiasing parameters $n, \alpha$.

**Output:** Predictions $\{\hat{\boldsymbol{y}}_{e,i}\}_{i=1}^M$.

1 Initialize $\tilde{\theta}_a \leftarrow \tilde{\theta}_0, \tilde{\theta}_e \leftarrow \tilde{\theta}_0$, moving average of entropy $e_m = 0$ $c_{bias} \leftarrow \mathbf{0}$;

2 **for** $\boldsymbol{x}_i \in \mathcal{D}_{test}$ **do**

3      Predict $\hat{\boldsymbol{y}}_{a,i} = f_a(\boldsymbol{x}_i; \theta_a, \rho), \hat{\boldsymbol{y}}_{e,i} = f_e(\boldsymbol{x}_i; \theta_e)$ and compute entropy $E_i = E(\hat{\boldsymbol{y}}_{a,i}; \theta_a)$;

4      **if** $E_i > \tau_{Ent}$ **then**

5          **continue**;

6      **end**

7      Compute weight $w(\hat{\boldsymbol{y}}_{e,i})$ and de-biased representation $\hat{\boldsymbol{y}}'_{e,i}$;

8      Compute $KL(\hat{\boldsymbol{y}}_{a,i} || \hat{\boldsymbol{y}}'_{e,i}; \theta_a)$ and total objective $\mathcal{L}_{tte}$:

$$\mathcal{L}_{tte}(\hat{\boldsymbol{y}}_{a,i}, \hat{\boldsymbol{y}}'_{e,i}; \theta_a) = E(\hat{\boldsymbol{y}}_{a,i}; \theta_a) + KL(\hat{\boldsymbol{y}}_{a,i} || \hat{\boldsymbol{y}}'_{e,i}; \theta_a)$$

9      Compute gradient $\nabla_{\tilde{\theta}_a} \mathcal{L}_{tte}(\hat{\boldsymbol{y}}_{a,i}, \hat{\boldsymbol{y}}'_{e,i}; \theta_a)$;

10      Compute $\hat{\epsilon}(\tilde{\theta})$ per Eq. (4);

11      Compute gradient approximation: $g = \nabla_{\tilde{\theta}_a} \mathcal{L}_{tte}(\hat{\boldsymbol{y}}_{a,i}, \hat{\boldsymbol{y}}'_{e,i}; \theta_a)|_{\theta + \hat{\epsilon}(\theta)}$;

12      Update $\tilde{\theta}_a \leftarrow \tilde{\theta}_a - \eta g$;

13      Compute momentum $m$ from Eq.2 and update ensemble parameters: $\tilde{\theta}_e \leftarrow m \cdot \tilde{\theta}_e + (1 - m) \cdot \tilde{\theta}_a$;

14      Compute first-order batch statistics $\mu_{\hat{\boldsymbol{y}}_{a,i}} = \frac{1}{N} \sum_{i=1}^N \hat{\boldsymbol{y}}_{a,i}$;

15      Update bias vector: $c_{bias} \leftarrow n \cdot c_{bias} + (1 - n) \cdot \mu_{\hat{\boldsymbol{y}}_{a,i}}$;

16      $e_m = 0.9 \times e_m + 0.1 \times E(\hat{\boldsymbol{y}}_{a,i}; \theta_a + \hat{\epsilon}(\theta_a))$ **if** $e_m \neq 0$ **else** $E(\boldsymbol{x}_i; \theta_a + \hat{\epsilon}(\theta_a))$;

17      **if** $e_m < e_0$ **then**

18          Recover model weights: $\tilde{\theta}_a \leftarrow \tilde{\theta}_0, \tilde{\theta}_e \leftarrow \tilde{\theta}_0$

19      **end**

20 **end**

---

---

**Algorithm 3:** TTE + DeYO (Lee et al., 2024)

---

**Input:** Test samples $\mathcal{D}_{\text{test}} = \{\boldsymbol{x}_i\}_{i=1}^M$, model $f_a(\cdot; \theta_a, \rho)$ with trainable parameters $\tilde{\theta}_a \subset \theta_a$ and a dropout ratio $\rho$,
  ensemble model $f_e(\cdot; \theta_e)$ with $\tilde{\theta}_e$ aligned to $\tilde{\theta}_a$, bias vector $\boldsymbol{c}_{bias}$, an object-destructive transformation $\mathcal{A}$, step
  size $\eta > 0$, and hyperparameters $\text{Ent}_0$, $\tau_{\text{Ent}}$, $\tau_{\text{PLPD}} > 0$, momentum $m$, debiasing parameters $n$, $\alpha$.
**Output:** Predictions $\{\hat{\boldsymbol{y}}_i\}_{i=1}^M$.

1 Initialize $\tilde{\theta}_a \leftarrow \tilde{\theta}_0$, $\tilde{\theta}_e \leftarrow \tilde{\theta}_0$, $c_{bias} \leftarrow \boldsymbol{0}$;
2 **for** $\boldsymbol{x}_i \in \mathcal{D}_{test}$ **do**
3   Predict $\hat{\boldsymbol{y}}_{a,i} = f_a(\boldsymbol{x}_i; \theta_a, \rho)$, $\hat{\boldsymbol{y}}_{e,i} = f_e(\boldsymbol{x}_i; \theta_e)$ and compute entropy $E_i = E(\hat{\boldsymbol{y}}_{a,i}; \theta_a)$;
4   **if** $E_i > \tau_{Ent}$ **then**
5    | continue;
6   **end**
7   Obtain $x'_i = \mathcal{A}(\boldsymbol{x}_i)$;
8   Compute pseudo-label probability difference $\text{PLPD}_\theta(\boldsymbol{x}_i, \boldsymbol{x}'_i)$;
9   **if** $PLPD_\theta(\boldsymbol{x}_i, \boldsymbol{x}'_i) < \tau_{PLPD}$ **then**
10    | continue;
11   **end**
12   Compute weight $w(\hat{\boldsymbol{y}}_{e,i})$ and de-biased representation $\hat{\boldsymbol{y}}'_{e,i}$;
13   Compute $KL(\hat{\boldsymbol{y}}_{a,i} || \hat{\boldsymbol{y}}'_{e,i}; \theta_a)$ and total objective $\mathcal{L}_{tte}$:

$$\mathcal{L}_{tte}(\hat{\boldsymbol{y}}_{a,i}, \hat{\boldsymbol{y}}'_{e,i}; \theta_a) = E(\hat{\boldsymbol{y}}_{a,i}; \theta_a) + KL(\hat{\boldsymbol{y}}_{a,i} || \hat{\boldsymbol{y}}'_{e,i}; \theta_a)$$

14   Compute sample weight $\alpha_{\theta_a}(\boldsymbol{x}_i)$;
15   Compute the overall loss $L_{\text{DeYO}} = \alpha_{\theta_a}(\boldsymbol{x}_i) \cdot \mathcal{L}_{tte}$;
16   Compute gradient $g = \nabla_{\tilde{\theta}_a} L_{\text{DeYO}}$;
17   Update $\tilde{\theta}_a \leftarrow \tilde{\theta}_a - \eta g$;
18   Compute momentum $m$ from Eq.2 and update ensemble parameters: $\tilde{\theta}_e \leftarrow m \cdot \tilde{\theta}_e + (1 - m) \cdot \tilde{\theta}_a$;
19   Compute first-order batch statistics $\mu_{\hat{\boldsymbol{y}}_{a,i}} = \frac{1}{N} \sum_{i=1}^N \hat{\boldsymbol{y}}_{a,i}$;
20   Update bias vector: $c_{bias} \leftarrow n \cdot c_{bias} + (1 - n) \cdot \mu_{\hat{\boldsymbol{y}}_{a,i}}$;
21 **end**

## C.3 COMPUTATIONAL COMPLEXITY

Integrating TTE required only an additional feedforward pass for $f_e$, resulting in minimal computational overhead. Table 9 details the computation and runtime required for adaptation with ViTBase under Gaussian noise (ImageNet-C, level 5). ViTBase was chosen for its superior performance over ResNet. Integrating TTE introduced an additional network, $f_e$, requiring only a single feedforward pass to compute $\hat{\boldsymbol{y}}_e$, adding minimal computational overhead. For example, a single feedforward pass without adaptation took 3 minutes 57 seconds. That was why TTE integration resulted in about a 4-minute increase in GPU time. In contrast to CoTTA, which relied on dense augmentations and multiple feedforward passes for knowledge distillation, TTE maintained efficiency by requiring only a single pass.

Table 9: Computational complexity and runtime for adaptation using ViTBase under Gaussian noise (ImageNet-C, level 5) with a Label Shifts setup. The total sample size is 100,000. **Bold** numbers present accuracy gain by applying TTE.

| Methods | #Model | #Forward | #Backward | Other Computation | GPU time (100k images) | Accuracy (%) |
|---|---|---|---|---|---|---|
| No adapt | 1 | 100k | - | n/a | 3 min 57 sec ($\times 1.0$) | 9.4 |
| Tent | 1 | 100k | 100k | n/a | 8 min 12 sec ($\times 2.1$) | 30.9 |
| SAR | 1 | 100k+75k | 75k+72k | weight perturbation | 14 min 46 sec ($\times 3.7$) | 46.6 |
| DeYO | 1 | 100k+92k | 75k | probability difference | 11 min 49 sec ($\times 3.0$) | 49.1 |
| CoTTA | 2 | 200k+479k | 100k | anchor probability | 81 min 14 sec ($\times 20.5$) | 34.1 |
| Tent+TTE | | 200k | 100k | | 12 min 30 sec ($\times 3.2$) | 46.6(**+15.7**) |
| SAR+TTE | 2 | 200k+72k | 72k+72k | $L_{rkl}$, $\boldsymbol{c}_{bias}$ | 20 min 30 sec ($\times 5.2$) | 48.6(**+2.0**) |
| DeYO+TTE | | 200k+89k | 72k | | 15 min 47 sec ($\times 4.0$) | 54.0(**+4.9**) |

# D    FURTHER ANALYSIS

## D.1    FAILURE CASE ANALYSIS

In this section, we analyzed failure cases when applying ensemble strategies to TTA alone. Two ensemble strategies were applied to a representative TTA method, Tent (Wang et al., 2021): constructing an ensemble network $f_e$ through a weight-space ensemble of the original TTA model $f_a$ and using knowledge distillation with standard KL divergence between outputs. Additionally, dropout was applied to the penultimate layer of $f_a$ to further enhance ensemble representations. The hyperparameters for ensemble strategies were set as $m_0 = 1.0$, $\tau = 1.0$ and a dropout ratio 0.6.

We evaluated classification accuracy under the challenging Label Shift scenario on ViTBase and ImageNet-C (level 5). To gain deeper insights, the loss barrier between $f_a$ and $f_e$ was measured as a typical metric for assessing linear mode connectivity, following Fort et al. (2020). The loss barrier was calculated along a linear interpolation path between the models in weight space, as described by

$$\max_{\alpha \in [0,1]} \left( \hat{R}(\theta_\alpha) - \frac{1}{2} \left( \hat{R}(\theta_e) + \hat{R}(\theta_a) \right) \right), \tag{7}$$

where $\hat{R}(\theta_\alpha) = \frac{1}{N} \sum_i^N L(f_\alpha(x_i; \theta_\alpha), y_i)$ and $\alpha$ is interpolation coefficient in $[0, 1]$. Here, $L$ is classification loss with input $x$ and label $y$.

Table 10 compares TTA with and without the two ensemble strategies. Figure 3 visualizes the average accuracy profile and prediction distributions for two representative datasets. The results show that TTA is unstable in the challenging scenario and can sometimes perform worse than no adaptation, due to biased class predictions (we call it model collapse). Applying ensemble strategies could occasionally worsen performance by constructing ensembles with unreliable $f_a$. Additional methods are needed to prevent the model from biasing.

Table 10: Classification accuracy (%) of ViTBase measured under Label Shifts (ImageNet-C, level 5). Red number indicates lower performance than without adaptation, signifying model collapse. **Bold number** highlights the best performance.

| | Gauss | Shot | Impul | Defoc | Glass | Motio | Zoom | Snow | Frost | Fog | Brit | Contr | Elast | Pixel | JPEG | Avg |
|---|---|---|---|---|---|---|---|---|---|---|---|---|---|---|---|---|
| No Adapt | 9.4 | 6.7 | 8.3 | 29.1 | 23.4 | 34.0 | 27.0 | 15.8 | 26.3 | 47.4 | 54.7 | 43.9 | 30.5 | 44.5 | 47.6 | 29.9 |
| TTA | 20.1 | 1.2 | 19.0 | 54.9 | 53.4 | 58.9 | 54.1 | 14.3 | 13.6 | 69.8 | 76.3 | 66.4 | 59.8 | 69.8 | 66.9 | 46.6 |
| TTA + Ensemble | **47.0** | **1.4** | **32.7** | **55.5** | **54.8** | **59.5** | **55.4** | **14.9** | **17.6** | **70.8** | **76.9** | **66.7** | **62.0** | **70.4** | **67.5** | **50.2** |

## D.2    MATHEMATICAL ANALYSIS OF REVERSE KL DIVERGENCE

In this section, we mathematically analyzed the robustness of reverse KL divergence in the context of TTA. Since KL divergence is asymmetric, swapping its inputs resulted in different effect. To understand this difference, we computed the gradients of both standard and reverse KL divergence when integrated with a typical TTA objective. Specifically, we used Tent (Wang et al., 2021), which minimized Shannon entropy via $-\hat{y}_a \log(\hat{y}_a)$. The objective $L_{tte}$ could be described in two ways as

$$L_{tte}(\hat{y}_a, \hat{y}_e') = \begin{cases} H(\hat{y}_a) + KL(\hat{y}_e' || \hat{y}_a) & \text{for Standard} \\ H(\hat{y}_a) + KL(\hat{y}_a || \hat{y}_e') & \text{for Reverse} \end{cases} \tag{8}$$

where $H(\hat{y}_a) = -\sum_c p(\hat{y}_a^c) \log p(\hat{y}_a^c)$ and $KL(p(\hat{y}_a^c) || p(\hat{y}_e'^c)) = -\sum_c p(\hat{y}_a^c) \log(p(\hat{y}_e'^c)/p(\hat{y}_a^c))$. Here, $p(\hat{y}_a^c)$ denotes the probability of class $c$ with the softmax function $p$. To simplify expressions, $p(\hat{y}_a^c)$ and $p(\hat{y}_e'^c)$ are denoted as $q_a^c$ and $q_e'^c$. The gradient magnitude of the objective function is calculated for the weight $\theta_a$ as

$$\left| \frac{\partial L_{tte}(\hat{y}_a, \hat{y}_e')}{\partial \theta_a} \right| = \begin{cases} \sum_c -(1 + \log q_a^c + q_e'^c/q_a^c) & \text{for Standard} \\ \sum_c -\log q_e'^c & \text{for Reverse} \end{cases} \tag{9}$$

In standard KL divergence, the term $q_e'^c/q_a^c$ produces larger gradients for samples where $q_e'^c$ is high but $q_a'^c$ is low. This term results in those misaligned samples being implicitly weighted more during gradient updates, as compared to samples where $q_e'^c$ and $q_a^c$ align. While this weighting can be beneficial when $q_e'^c$ is reliable, it may lead to overfitting to noise if $q_e'^c$ is noisy. In contrast, reverse KL divergence lacks the $q_e'^c/q_a^c$ term, treating all samples only based on ensemble prediction ($-\sum_c \log q_e'^c$). Figure 4 empirically supports the analysis by comparing classification accuracy and gradient ratio during TTA process. Reverse KL divergence reduces the gradient weight on incorrect predictions while increasing it for correct ones.

## D.3 FURTHER EXPERIMENTS FOR TENT+TTE AND SAR+TTE

To further validate the robustness of TTE, we revisited the experiments in Tables 3 and 4, incorporating other baselines integrated with TTE (e.g., Tent+TTE and SAR+TTE). Tables 11 and 12 present the additional results alongside some previous findings, demonstrating consistency with the DeYO+TTE results and reaffirming the effectiveness of TTE.

Table 11: Continual TTA with non i.i.d. conditions. Classification accuracy (%) with ImageNet-C (level 5). Underline depicts performance improvement when applying TTE. **Bold** numbers are the best results.

| | Adaptation Order → | | | | | | | | | | | | | | | |
|---|---|---|---|---|---|---|---|---|---|---|---|---|---|---|---|---|
| | Gauss | Shot | Impul | Defoc | Glass | Motion | Zoom | Snow | Frost | Fog | Brit | Contr | Elastic | Pixel | JPEG | Avg |
| ResNet50-GN | 18.0 | 19.8 | 17.9 | 19.8 | 11.4 | 21.4 | 24.9 | 40.4 | 47.3 | 33.6 | 69.3 | 36.3 | 18.6 | 28.4 | 52.3 | 30.6 |
| • Tent | 3.9 | 1.8 | 1.6 | 0.1 | 0.1 | 0.1 | 0.1 | 0.1 | 0.1 | 0.1 | 0.1 | 0.1 | 0.1 | 0.1 | 0.1 | 0.6 |
| • Tent + TTE | 36.2 | 27.9 | 46.8 | 48.1 | 42.1 | 30.4 | 49.4 | 45.2 | 36.8 | 34.7 | 50.3 | 52.8 | 39.4 | 67.0 | 55.9 | 44.2 |
| • SAR | 33.1 | 16.8 | 44.7 | 44.6 | 42.3 | 18.8 | 45.7 | 37.8 | 39.7 | 9.3 | 3.1 | 2.1 | 0.7 | 5.4 | 1.2 | 23.0 |
| • SAR + TTE | 35.9 | 28.3 | 46.9 | 48.9 | 43.1 | 30.7 | 50.1 | 45.6 | 38.2 | 35.5 | 51.5 | 54.4 | 39.7 | 67.8 | 57.0 | 44.9 |
| • DeYO | 28.1 | 3.7 | 7.2 | 0.9 | 0.1 | 0.1 | 0.1 | 0.1 | 0.1 | 0.1 | 0.1 | 0.1 | 0.1 | 0.1 | 0.1 | 2.8 |
| • DeYO + TTE | **43.0** | **31.7** | **52.7** | **52.3** | **45.1** | **36.4** | **52.2** | **53.6** | **40.9** | **43.0** | **59.1** | **60.3** | **47.4** | **68.9** | **59.0** | **49.7** |
| ViTBase | 9.4 | 43.9 | 30.5 | 44.5 | 29.1 | 6.7 | 8.3 | 27.0 | 15.8 | 23.4 | 34.0 | 54.7 | 26.3 | 47.4 | 47.6 | 29.9 |
| • Tent | 30.9 | 18.5 | 7.1 | 0.1 | 0.1 | 0.1 | 0.1 | 0.1 | 0.1 | 0.1 | 0.1 | 0.1 | 0.1 | 0.2 | 0.1 | 3.9 |
| • Tent + TTE | 46.6 | 51.6 | 61.0 | 59.9 | 49.5 | 51.3 | 60.8 | 62.8 | 49.4 | 55.2 | 64.5 | 68.4 | 52.5 | 74.6 | 66.1 | 58.3 |
| • SAR | 46.6 | 54.2 | 54.7 | 49.0 | 38.7 | 38.9 | 45.3 | 45.3 | 37.5 | 40.0 | 45.8 | 50.0 | 37.3 | 58.6 | 48.6 | 46.0 |
| • SAR + TTE | 48.6 | 53.8 | 62.3 | **62.8** | 51.5 | 54.2 | 62.2 | 64.8 | 51.5 | 57.7 | 66.8 | 69.8 | 55.1 | **75.3** | 67.5 | 60.3 |
| • DeYO | 49.1 | 36.7 | 63.3 | 61.4 | 52.8 | 49.9 | 59.1 | 62.2 | 48.9 | 54.0 | 61.5 | 69.2 | 2.2 | 64.8 | 67.2 | 53.5 |
| • DeYO + TTE | **54.0** | **55.2** | **64.6** | 62.5 | **54.2** | **55.9** | **62.6** | **66.8** | **52.0** | **59.1** | **68.5** | **69.9** | **55.2** | 74.9 | **68.4** | **61.6** |

Table 12: TTA with natural distribution shifts. Classification accuracy (%) with ImageNet-Sketch (S), ImageNet-Rendition (R), ImageNet-V2 (V2). Underline depicts performance improvement when applying TTE. **Bold** numbers are the best results.

(a) Label Shifts

| Methods | S | R | V2 | Avg |
|---|---|---|---|---|
| ResNet50-GN | 29.2 | 40.8 | 68.9 | 46.3 |
| • Tent | 30.8 | 41.5 | **68.9** | 47.1 |
| • Tent+TTE | 33.4 | 42.1 | **68.9** | 48.1 |
| • SAR | 30.6 | 41.6 | **68.9** | 47.0 |
| • SAR+TTE | 32.5 | 41.9 | **68.9** | 47.8 |
| • DeYO | 34.6 | 44.4 | **68.9** | 49.3 |
| • DeYO+TTE | **36.9** | **45.1** | **68.9** | **50.3** |
| ViTBase | 18.2 | 43.1 | 66.2 | 37.9 |
| • Tent | 8.8 | 41.9 | 68.9 | 39.8 |
| • Tent+TTE | 36.9 | 52.1 | 69.1 | 52.7 |
| • SAR | 17.8 | 45.1 | 68.5 | 43.8 |
| • SAR+TTE | 37.0 | 51.9 | 68.8 | 52.6 |
| • DeYO | 42.4 | 58.6 | **71.1** | 57.3 |
| • DeYO+TTE | **43.6** | **59.1** | **71.1** | **57.9** |

(b) Batch Size 1

| Methods | S | R | V2 | Avg |
|---|---|---|---|---|
| ResNet50-GN | 29.2 | 40.8 | 68.9 | 46.3 |
| • Tent | 31.9 | 42.1 | 68.9 | 47.6 |
| • Tent+TTE | 36.2 | 44.1 | 68.9 | 49.7 |
| • SAR | 31.6 | 41.9 | 68.9 | 47.5 |
| • SAR+TTE | 31.7 | 41.8 | 68.9 | 47.5 |
| • DeYO | 37.0 | 46.0 | 69.0 | 50.7 |
| • DeYO+TTE | **39.1** | **47.1** | 69.0 | **51.7** |
| ViTBase | 18.2 | 43.1 | 66.2 | 37.9 |
| • Tent | 7.0 | 40.6 | 69.1 | 38.9 |
| • Tent+TTE | 38.5 | 53.7 | 69.3 | 53.8 |
| • SAR | 26.4 | 44.7 | 69.4 | 46.8 |
| • SAR+TTE | 33.7 | 50.4 | 68.9 | 51.0 |
| • DeYO | 43.6 | 59.8 | 71.2 | 58.2 |
| • DeYO+TTE | **45.0** | **61.3** | **71.3** | **59.2** |

## D.4 FURTHER EXPERIMENTS FOR CATASTROPHIC FORGETTING

The proposed TTE effectively prevents model collapse and demonstrates stable optimization across extensive TTA experiments. To further evaluate its stability, we conducted additional experiments to determine whether TTE can address catastrophic forgetting—a phenomenon where a model exhibits severe performance degradation on source domain dataset after adaptation. To test this, we concurrently measured the accuracy on the clean ImageNet dataset immediately after each adaptation to a distribution in Table 11. Figure 8 shows that integrating TTE successfully mitigates forgetting issues, whereas other baseline methods suffer from them.

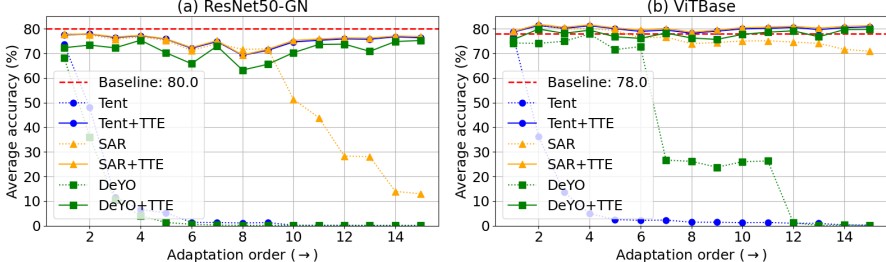

Figure 8: Comparison of preventing catastrophic forgetting on Continual TTA with non-i.i.d. conditions. Classification accuracy on in-distribution data (ImageNet). Each point was measured after TTA on each out-of-distribution data in Table 11.

## D.5 Further experiments for weight averaging strategy

The proposed ensemble strategy is inspired by weight-averaging methods originally developed for offline domain generalization. In this section, we compare the proposed adaptive weight-averaging method with the stochastic weight-averaging (SWA) method (Izmailov et al., 2018). For this comparison, we create a variant of the TTE method by replacing the proposed ensemble strategy with SWA while keeping all other components identical. SWA focuses on the uniform averaging of TTA models generated during the TTA process and is implemented as follows:

$$w_{\text{swa}} \leftarrow \frac{w_{\text{swa}} \cdot n_{\text{models}} + w}{n_{\text{models}} + 1} \tag{10}$$

where $n_{\text{models}}$ represents the number of ensemble models, $w_{\text{swa}}$ and $w$ denote the parameters of the TTA model $f_a$ and the ensemble model $f_e$, respectively. The value of $n_{\text{models}}$ is determined by the frequency $f$ of model selection during TTA. We extensively optimized $f$ within the range [1, 20] and selected 1 for ResNet50-GN and 5 for ViTBase to achieve maximum performance. Table 13 presents the comparison results on ImageNet-C under the Label Shifts setup. The results indicate that the proposed adaptive scheme achieves higher adaptation performance compared to the SWA approach.

Table 13: Comparison study between different weight averaging schemes. Classification accuracy (%) with the Label Shifts setup (ImageNet-C, level 5). Note that swa stands for stochastic weight averaging. **Bold** numbers are the best results.

| Label Shifts | Gauss | Shot | Impul | Defoc | Glass | Motion | Zoom | Snow | Frost | Fog | Brit | Contr | Elastic | Pixel | JPEG | Avg |
|---|---|---|---|---|---|---|---|---|---|---|---|---|---|---|---|---|
| ResNet50-GN | 18.0 | 19.8 | 17.9 | 19.8 | 11.4 | 21.4 | 24.9 | 40.4 | 47.3 | 33.6 | 69.3 | 36.3 | 18.6 | 28.4 | 52.3 | 30.6 |
| DeYO | 28.1 | 44.3 | 42.9 | 23.4 | 16.6 | 41.5 | 6.1 | 52.9 | 52.0 | 20.2 | 73.2 | 53.0 | 37.7 | 60.0 | 59.4 | 40.8 |
| + TTE (swa) | 39.8 | 42.9 | 41.6 | 30.2 | 28.7 | 39.8 | 40.4 | 52.9 | 50.9 | 58.4 | 73.3 | 51.6 | 45.1 | 57.5 | 58.5 | 47.4 |
| + TTE (ours) | **43.0** | **45.3** | **44.0** | **34.4** | **33.6** | **43.4** | **46.0** | **55.2** | **53.6** | **61.0** | **73.4** | **54.4** | **51.5** | **61.0** | **60.2** | **50.7** |
| ViTBase | 9.4 | 6.7 | 8.3 | 29.1 | 23.4 | 34.0 | 27.0 | 15.8 | 26.3 | 47.4 | 54.7 | 43.9 | 30.5 | 44.5 | 47.6 | 29.9 |
| DeYO | 49.1 | 35.9 | 53.6 | 57.6 | 58.6 | 63.8 | 37.5 | 67.9 | 66.0 | 73.1 | 77.9 | 66.5 | 68.6 | 73.5 | 70.1 | 61.3 |
| + TTE (swa) | 53.5 | 53.6 | 54.4 | 58.7 | 59.4 | 63.7 | 60.7 | 67.9 | 66.3 | **73.9** | **78.3** | **68.4** | 68.7 | 73.6 | 70.2 | 64.7 |
| + TTE (ours) | **54.0** | **54.7** | **55.2** | **58.9** | **59.7** | **64.5** | **62.1** | **68.3** | **66.7** | **73.9** | 78.0 | 68.3 | **69.3** | **73.8** | **70.3** | **65.2** |

## D.6 Further experiments for long-range true-biased scenarios

The proposed de-biasing scheme is designed to prevent the model from biasing towards specific classes, a key issue when models collapse. However, in cases where true label stream is heavily biased (i.e., a model encounter the same class samples consecutively for over 100 iterations), we assessed whether the de-biasing scheme could still perform effectively. To simulate extreme scenarios, we combined a batch size of 1 with the Label Shifts setup, where class-imbalanced streams persisted over 100 iterations.

Figure 9 shows the average accuracy across all distributions in the Label Shifts setup for ViTBase. Unfortunately, the default de-biasing configuration ($\alpha = 3.0$, $n = 0.99$) penalized true-biased predictions, providing lower accuracy than the baseline DeYO. By reducing the de-biasing strength

($\alpha = 1.0$) and more gradually estimating model bias ($n = 0.999$), TTE stabilized and achieved higher accuracy than DeYO. However, over-tuning hyperparameter based on test data performance could be controversial. Future work should explore additional methods to distinguish between true bias and false bias, which can lead to model collapse.

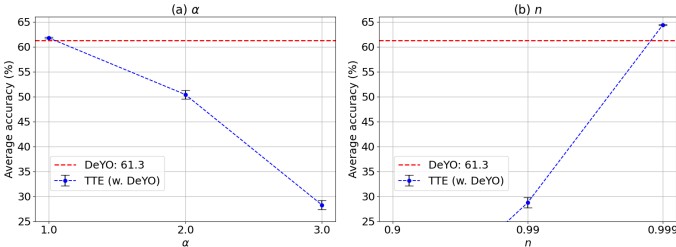

Figure 9: Average classification accuracy (%) in varing with hyperparameter of de-biasing schemes in Lable Shifts for ViTBase (ImageNet-C, level 5). The extremely biased scenario is used to assume that same class samples are fed into the model consecutively over 100 times.

# E    MAIN RESULTS WITH STANDARD DEVIATION

In this paper, experiments were conducted with three different random seeds: 2022, 2023, and 2024. The results presented in Tables 1, 2, and 3 include the mean values across these three repetitions. The corresponding standard deviations are provided in Tables 14, 15, and 16, respectively.

Table 14: Integration with previous TTA approaches. Classification accuracy (%) with Label Shifts and Batch Size 1 setups (ImageNet-C, level 5). Underline depicts performance improvement when applying TTE. **Bold** numbers are the best results.

| Label Shifts | Gauss | Shot | Impul | Defoc | Glass | Motion | Zoom | Snow | Frost | Fog | Brit | Contr | Elastic | Pixel | JPEG | Avg |
|---|---|---|---|---|---|---|---|---|---|---|---|---|---|---|---|---|
| ResNet50-GN | 18.0 | 19.8 | 17.9 | 19.8 | 11.4 | 21.4 | 24.9 | 40.4 | 47.3 | 33.6 | 69.3 | 36.3 | 18.6 | 28.4 | 52.3 | 30.6 |
| • Tent | 3.9 ±0.27 | 4.6 ±0.5 | 4.7 ±0.24 | 16.4 ±0.04 | 6.0 ±2.16 | 27.2 ±0.6 | 29.0 ±1.94 | 18.9 ±1.0 | 27.4 ±0.51 | 2.4 ±0.25 | 72.1 ±0.03 | 46.1 ±0.2 | 8.1 ±0.76 | 52.4 ±0.03 | 56.1 ±0.07 | 25.0 ±0.48 |
| • Tent + TTE | 36.2 ±0.23 | 38.4 ±0.11 | 37.3 ±0.05 | 29.6 ±0.1 | 25.8 ±0.14 | 36.1 ±0.14 | 38.6 ±0.22 | 50.2 ±0.23 | 48.8 ±0.06 | 54.9 ±0.14 | 72.1 ±0.01 | 47.7 ±0.16 | 40.1 ±0.07 | 53.0 ±0.13 | 56.4 ±0.02 | 44.4 ±0.06 |
| • SAR | 33.1 ±0.79 | 36.5 ±0.17 | 35.2 ±0.72 | 18.9 ±1.19 | 20.8 ±0.41 | 33.3 ±3.78 | 29.8 ±6.04 | 27.8 ±0.36 | 44.9 ±22.21 | 35.2 ±0.05 | 71.9 ±0.18 | 46.6 ±2.24 | 7.6 ±0.11 | 52.1 ±0.09 | 56.2 ±1.04 | 36.7 |
| • SAR + TTE | 35.9 ±0.25 | 38.4 ±0.29 | 37.3 ±0.18 | 29.7 ±0.32 | 25.3 ±0.4 | 36.2 ±0.1 | 37.4 ±0.16 | 49.8 ±0.19 | 48.0 ±0.18 | 53.2 ±0.53 | 71.9 ±0.06 | 47.6 ±0.21 | 39.1 ±0.29 | 52.8 ±0.04 | 56.2 ±0.09 | 43.9 ±0.09 |
| • DeYo | 28.1 ±18.89 | 44.3 ±0.17 | 42.9 ±0.42 | 23.4 ±0.73 | 16.6 ±10.19 | 41.5 ±0.34 | 6.1 ±1.18 | 52.9 ±0.46 | 52.0 ±0.19 | 20.2 ±27.32 | 73.2 ±0.08 | 53.0 ±0.25 | 37.7 ±14.93 | 60.0 ±0.1 | 59.4 ±0.05 | 40.8 ±0.87 |
| • DeYo + TTE | 43.0 ±0.31 | 45.3 ±0.27 | 44.0 ±0.14 | 34.4 ±0.17 | 33.6 ±0.38 | 43.4 ±0.28 | 46.0 ±0.21 | 55.2 ±0.18 | 53.6 ±0.18 | 61.0 ±0.31 | 73.4 ±0.18 | 54.4 ±0.09 | 51.5 ±0.45 | 61.0 ±0.04 | 60.2 ±0.09 | 50.7 ±0.13 |
| ViTBase | 9.4 | 6.7 | 8.3 | 29.1 | 23.4 | 34.0 | 27.0 | 15.8 | 26.3 | 47.4 | 54.7 | 43.9 | 30.5 | 44.5 | 47.6 | 29.9 |
| • Tent | 30.9 ±11.44 | 1.0 ±0.15 | 23.2 ±6.06 | 54.9 ±0.04 | 53.2 ±0.27 | 58.8 ±0.13 | 54.3 ±0.2 | 13.3 ±0.91 | 12.5 ±0.86 | 69.8 ±0.03 | 76.3 ±0.06 | 66.3 ±0.3 | 59.7 ±0.06 | 69.8 ±0.09 | 66.8 ±0.09 | 47.4 ±1.01 |
| • Tent + TTE | 46.6 ±0.33 | 45.4 ±0.09 | 47.9 ±0.07 | 55.4 ±0.05 | 54.4 ±0.19 | 59.0 ±0.08 | 55.3 ±0.13 | 62.7 ±0.08 | 62.0 ±0.14 | 69.9 ±0.14 | 76.4 ±0.15 | 66.3 ±0.16 | 61.7 ±0.23 | 69.8 ±0.06 | 67.0 ±0.05 | 60.0 ±0.05 |
| • SAR | 46.6 ±2.46 | 29.5 ±14.73 | 48.1 ±1.37 | 55.2 ±0.07 | 54.2 ±0.06 | 59.0 ±0.08 | 54.6 ±0.31 | 58.0 ±5.7 | 44.1 ±2.78 | 69.8 ±0.14 | 76.2 ±0.14 | 66.1 ±0.18 | 60.9 ±0.06 | 69.7 ±0.06 | 66.6 ±0.08 | 57.2 ±0.91 |
| • SAR + TTE | 48.6 ±0.42 | 46.4 ±1.87 | 48.4 ±1.65 | 55.8 ±0.05 | 55.2 ±0.22 | 59.4 ±0.12 | 55.8 ±0.17 | 63.2 ±0.12 | 62.4 ±0.17 | 70.1 ±0.1 | 76.4 ±0.1 | 66.4 ±0.22 | 62.4 ±0.18 | 70.0 ±0.07 | 67.0 ±0.09 | 60.5 ±0.24 |
| • DeYo | 49.1 ±5.79 | 35.9 ±25.22 | 53.6 ±0.3 | 57.6 ±0.11 | 58.6 ±0.07 | 63.8 ±21.33 | 37.5 ±0.19 | 67.9 ±0.06 | 66.0 ±0.04 | 73.1 ±0.09 | 77.9 ±0.04 | 66.5 ±0.06 | 68.6 ±0.09 | 73.5 ±0.09 | 70.1 ±0.05 | 61.3 ±3.5 |
| • DeYo + TTE | 54.0 ±0.37 | 54.7 ±0.49 | 55.2 ±0.14 | 58.9 ±0.2 | 59.7 ±0.06 | 64.5 ±0.05 | 62.1 ±0.58 | 68.3 ±0.13 | 66.7 ±0.06 | 73.9 ±0.2 | 78.0 ±0.04 | 68.3 ±0.26 | 69.3 ±0.1 | 73.8 ±0.16 | 70.3 ±0.04 | 65.2 ±0.11 |

| BS1 | Gauss | Shot | Impul | Defoc | Glass | Motion | Zoom | Snow | Frost | Fog | Brit | Contr | Elastic | Pixel | JPEG | Avg |
|---|---|---|---|---|---|---|---|---|---|---|---|---|---|---|---|---|
| ResNet50-GN | 18.0 | 19.8 | 17.9 | 19.8 | 11.4 | 21.4 | 24.9 | 40.4 | 47.3 | 33.6 | 69.3 | 36.3 | 18.6 | 28.4 | 52.3 | 30.6 |
| • Tent | 3.1 ±0.01 | 4.1 ±0.4 | 3.7 ±0.28 | 16.6 ±0.06 | 5.2 ±1.99 | 27.2 ±0.68 | 29.0 ±2.55 | 17.7 ±0.56 | 25.1 ±1.12 | 1.9 ±0.09 | 72.0 ±0.06 | 46.2 ±0.05 | 8.1 ±0.07 | 52.7 ±0.05 | 56.3 ±0.07 | 24.6 ±0.05 |
| • Tent + TTE | 41.6 ±0.03 | 43.9 ±0.1 | 42.7 ±0.03 | 33.8 ±0.07 | 31.2 ±0.05 | 41.0 ±0.06 | 44.1 ±0.13 | 53.5 ±0.08 | 52.2 ±0.11 | 59.3 ±0.11 | 73.1 ±0.06 | 51.3 ±0.16 | 47.8 ±0.14 | 57.7 ±0.17 | 58.1 ±0.03 | 48.7 ±0.01 |
| • SAR | 23.4 ±0.32 | 26.5 ±0.22 | 23.9 ±0.24 | 18.4 ±0.18 | 15.1 ±0.1 | 28.6 ±0.18 | 30.3 ±0.09 | 44.4 ±0.4 | 44.8 ±0.15 | 27.4 ±2.17 | 72.3 ±0.13 | 44.7 ±0.07 | 14.6 ±0.06 | 47.0 ±0.06 | 56.1 ±0.08 | 34.5 ±0.08 |
| • SAR + TTE | 25.9 ±0.07 | 28.6 ±0.21 | 26.7 ±0.25 | 23.7 ±0.07 | 17.7 ±0.2 | 30.8 ±0.1 | 32.4 ±0.2 | 48.0 ±0.13 | 46.1 ±0.08 | 42.1 ±0.97 | 72.2 ±0.04 | 45.2 ±0.08 | 34.2 ±0.18 | 47.7 ±0.14 | 56.1 ±0.04 | 38.5 ±0.08 |
| • DeYo | 41.3 ±0.23 | 44.2 ±0.1 | 42.4 ±0.18 | 23.7 ±0.33 | 25.1 ±0.18 | 41.4 ±0.09 | 19.9 ±9.34 | 54.6 ±0.04 | 52.2 ±0.07 | 1.9 ±0.12 | 73.4 ±0.12 | 53.4 ±0.11 | 39.9 ±10.57 | 59.9 ±0.23 | 59.7 ±0.09 | 42.2 ±1.19 |
| • DeYo + TTE | 42.5 ±0.09 | 44.9 ±0.28 | 43.5 ±0.24 | 34.8 ±0.13 | 32.8 ±0.22 | 43.3 ±0.19 | 45.9 ±0.05 | 55.8 ±0.19 | 53.7 ±0.12 | 60.5 ±0.12 | 73.4 ±0.05 | 54.4 ±0.04 | 51.0 ±0.3 | 60.9 ±0.04 | 60.4 ±0.18 | 50.5 ±0.06 |
| ViTBase | 9.4 | 6.7 | 8.3 | 29.1 | 23.4 | 34.0 | 27.0 | 15.8 | 26.3 | 47.4 | 54.7 | 43.9 | 30.5 | 44.5 | 47.6 | 29.9 |
| • Tent | 43.2 ±0.08 | 1.6 ±0.19 | 44.0 ±0.22 | 52.6 ±0.09 | 48.9 ±0.1 | 55.8 ±0.11 | 51.2 ±0.08 | 22.3 ±0.73 | 21.5 ±0.7 | 67.0 ±0.06 | 75.0 ±0.04 | 64.9 ±0.04 | 54.3 ±0.43 | 67.2 ±0.06 | 64.4 ±0.0 | 48.9 ±0.11 |
| • Tent + TTE | 49.2 ±0.06 | 48.8 ±0.08 | 50.1 ±0.05 | 55.8 ±0.07 | 55.8 ±0.07 | 60.2 ±0.08 | 56.9 ±0.1 | 64.3 ±0.12 | 63.6 ±0.13 | 71.4 ±0.03 | 76.9 ±0.07 | 67.0 ±0.08 | 64.1 ±0.08 | 70.7 ±0.01 | 68.1 ±0.03 | 61.6 ±0.01 |
| • SAR | 40.9 ±0.23 | 36.6 ±0.26 | 41.9 ±0.12 | 53.4 ±0.18 | 50.5 ±0.14 | 57.4 ±0.04 | 52.9 ±0.02 | 59.1 ±0.48 | 57.2 ±2.03 | 68.9 ±0.01 | 75.5 ±0.39 | 65.6 ±0.03 | 58.1 ±0.13 | 68.9 ±0.08 | 65.9 ±0.11 | 56.9 ±0.19 |
| • SAR + TTE | 43.6 ±0.48 | 40.4 ±0.24 | 44.3 ±0.28 | 55.2 ±0.15 | 53.1 ±0.17 | 59.2 ±0.04 | 55.4 ±0.17 | 61.5 ±0.39 | 61.9 ±0.1 | 70.7 ±0.08 | 76.7 ±0.03 | 66.7 ±0.08 | 61.9 ±0.15 | 70.2 ±0.1 | 67.3 ±0.09 | 59.2 ±0.01 |
| • DeYo | 53.1 ±0.1 | 51.2 ±3.1 | 54.3 ±0.15 | 58.8 ±0.14 | 59.6 ±0.09 | 64.0 ±0.17 | 37.4 ±6.68 | 68.1 ±0.1 | 66.4 ±0.12 | 73.7 ±0.16 | 78.3 ±0.06 | 68.2 ±0.05 | 68.5 ±0.17 | 73.7 ±0.04 | 70.5 ±0.05 | 63.1 ±0.33 |
| • DeYo + TTE | 53.8 ±0.14 | 54.0 ±0.07 | 54.6 ±0.16 | 59.1 ±0.01 | 59.7 ±0.09 | 64.4 ±0.07 | 62.3 ±0.16 | 68.4 ±0.11 | 66.8 ±0.15 | 73.9 ±0.13 | 78.3 ±0.03 | 68.5 ±0.08 | 69.2 ±0.11 | 73.9 ±0.04 | 70.7 ±0.03 | 65.2 ±0.0 |

Table 15: Integration with previous TTA approaches. Classification accuracy (%) with the Mix Shifts setup (ImageNet-C, level 5). Underline depicts performance improvement when applying TTE. **Bold** numbers are the best results.

| Methods | ResNet50-GN | ViTBase |
|---|---|---|
| No Adapt | 30.6 | 29.9 |
| • Tent | 33.1±0.12 | 52.3±3.99 |
| • Tent+TTE | 38.7±0.32 | 57.2±0.37 |
| • SAR | 38.1±0.21 | 57.1±0.04 |
| • SAR+TTE | 39.1±0.33 | 57.5±0.34 |
| • DeYO | 33.8±1.6 | 58.6±0.13 |
| • DeYO+TTE | **42.9**±0.18 | **60.6**±0.45 |

Table 16: Continual TTA with correlatively sampling. Classification accuracy (%) with ImageNet-C (level 5). **Bold** numbers are the best results.

| | Gauss | Shot | Impul | Defoc | Glass | Motion | Zoom | Snow | Frost | Fog | Brit | Contr | Elastic | Pixel | JPEG | Avg |
|---|---|---|---|---|---|---|---|---|---|---|---|---|---|---|---|---|
| | Correlative Sampling in Both of Domain and Class (Adaptation Order →) | | | | | | | | | | | | | | | |
| ResNet50-GN | 18.0 | 19.8 | 17.9 | 19.8 | 11.4 | 21.4 | 24.9 | 40.4 | 47.3 | 33.6 | 69.3 | 36.3 | 18.6 | 28.4 | 52.3 | 30.6 |
| ● Tent | 3.9 ±0.27 | 1.8 ±0.28 | 1.6 ±0.19 | 0.1 ±0.03 | 0.1 ±0.0 | 0.1 ±0.0 | 0.1 ±0.0 | 0.1 ±0.0 | 0.1 ±0.0 | 0.1 ±0.0 | 0.1 ±0.0 | 0.1 ±0.0 | 0.1 ±0.0 | 0.1 ±0.0 | 0.1 ±0.0 | 0.6 ±0.05 |
| ● CoTTA | 23.5 ±0.31 | 5.5 ±1.29 | 2.0 ±0.47 | 0.7 ±0.45 | 0.4 ±0.19 | 0.2 ±0.02 | 0.2 ±0.02 | 0.1 ±0.03 | 0.1 ±0.03 | 0.1 ±0.03 | 0.1 ±0.04 | 0.1 ±0.03 | 0.1 ±0.03 | 0.1 ±0.03 | 0.1 ±0.03 | 2.2 ±0.1 |
| ● SAR | 33.1 ±0.79 | 16.8 ±0.36 | 44.7 ±0.04 | 44.6 ±0.25 | 42.3 ±0.08 | 18.8 ±0.58 | 45.7 ±0.37 | 37.8 ±0.51 | 39.7 ±0.19 | 9.3 ±1.97 | 3.1 ±0.15 | 2.1 ±0.27 | 0.7 ±0.06 | 5.4 ±0.48 | 1.2 ±0.46 | 23.0 ±0.06 |
| ● DeYO | 28.1 ±18.89 | 3.7 ±2.67 | 7.2 ±8.28 | 0.9 ±0.94 | 0.1 ±0.03 | 0.1 ±0.03 | 0.1 ±0.03 | 0.1 ±0.03 | 0.1 ±0.02 | 0.1 ±0.03 | 0.1 ±0.01 | 0.1 ±0.02 | 0.1 ±0.01 | 0.1 ±0.02 | 0.1 ±0.03 | 2.8 ±1.86 |
| ● TTE (w. DeYO) | **43.0** ±0.31 | **31.7** ±0.39 | **52.7** ±0.2 | **52.3** ±0.12 | **45.1** ±0.24 | **36.4** ±0.16 | **52.2** ±0.22 | **53.6** ±0.11 | **40.9** ±0.19 | **43.0** ±0.25 | **59.1** ±0.06 | **60.3** ±0.27 | **47.4** ±0.23 | **68.9** ±0.2 | **59.0** ±0.2 | **49.7** ±0.07 |
| ViTBase | 9.4 | 43.9 | 30.5 | 44.5 | 29.1 | 6.7 | 8.3 | 27.0 | 15.8 | 23.4 | 34.0 | 54.7 | 26.3 | 47.4 | 47.6 | 29.9 |
| ● Tent | 30.9 ±11.44 | 18.5 ±24.76 | 7.1 ±9.74 | 0.1 ±0.0 | 0.1 ±0.04 | 0.1 ±0.0 | 0.1 ±0.0 | 0.1 ±0.01 | 0.1 ±0.0 | 0.1 ±0.0 | 0.1 ±0.0 | 0.1 ±0.0 | 0.1 ±0.0 | 0.2 ±0.05 | 0.1 ±0.0 | 3.9 ±3.06 |
| ● CoTTA | 34.1 ±2.79 | 13.6 ±9.93 | 2.2 ±2.9 | 0.1 ±0.0 | 0.1 ±0.0 | 0.1 ±0.0 | 0.1 ±0.0 | 0.1 ±0.0 | 0.1 ±0.0 | 0.1 ±0.0 | 0.1 ±0.0 | 0.1 ±0.0 | 0.1 ±0.0 | 0.1 ±0.0 | 0.1 ±0.0 | 3.4 ±0.93 |
| ● SAR | 46.6 ±2.46 | 54.2 ±0.32 | 54.7 ±5.12 | 49.0 ±21.9 | 38.7 ±17.14 | 38.9 ±22.57 | 45.3 ±23.45 | 45.3 ±28.06 | 37.5 ±20.71 | 40.0 ±26.08 | 45.8 ±31.59 | 50.0 ±27.96 | 37.3 ±25.39 | 58.6 ±24.07 | 48.6 ±26.9 | 46.0 ±19.67 |
| ● DeYO | 49.1 ±5.79 | 36.7 ±25.19 | 63.3 ±3.21 | 61.4 ±0.43 | 52.8 ±0.25 | 49.9 ±3.17 | 59.1 ±2.02 | 62.2 ±3.96 | 48.9 ±2.31 | 54.0 ±1.38 | 61.5 ±4.79 | 69.2 ±0.71 | 2.2 ±1.4 | 64.8 ±9.61 | 67.2 ±0.5 | 53.5 ±2.09 |
| ● TTE (w. DeYO) | **54.0** ±0.37 | **55.2** ±0.56 | **64.6** ±0.03 | **62.5** ±0.31 | **54.2** ±0.06 | **55.9** ±0.22 | **62.6** ±0.02 | **66.8** ±0.32 | **52.0** ±0.94 | **59.1** ±0.8 | **68.5** ±0.5 | **69.9** ±0.45 | **55.2** ±1.1 | **74.9** ±0.1 | **68.4** ±0.21 | **61.6** ±0.36 |

Table 17: TTA with natural distribution shifts. Classification accuracy (%) with ImageNet-Sketch (S), ImageNet-Rendition (R), ImageNet-V2 (V2). **Bold** numbers are the best results.

(a) Label Shifts

| Methods | S | R | V2 | Avg |
|---|---|---|---|---|
| ResNet50-GN | 29.2 | 40.8 | 68.9 | 46.3 |
| ● Tent | 30.8±0.0 | 41.5±0.1 | **68.9**±0.0 | 47.1±0.0 |
| ● CoTTA | 30.8±0.1 | 41.2±0.1 | 68.8±0.1 | 46.9±0.0 |
| ● SAR | 30.6±0.1 | 41.6±0.0 | **68.9**±0.0 | 47.0±0.1 |
| ● DeYO | 34.6±0.4 | 44.4±0.2 | **68.9**±0.1 | 49.3±0.2 |
| ● TTE (w. DeYO) | **36.9**±0.0 | **45.1**±0.4 | **68.9**±0.1 | **50.3**±0.1 |
| ViTBase | 18.2 | 43.1 | 66.2 | 37.9 |
| ● Tent | 8.8±0.5 | 41.9±0.6 | 68.9±0.0 | 39.8±0.3 |
| ● CoTTA | 28.4±1.2 | 45.1±0.8 | 67.5±0.1 | 47.0±0.4 |
| ● SAR | 17.8±2.7 | 45.1±0.9 | 68.5±0.1 | 43.8±1.0 |
| ● DeYO | 42.4±0.6 | 58.6±0.3 | 71.1±0.2 | 57.3±0.2 |
| ● TTE (w. DeYO) | **43.6**±0.1 | **59.1**±0.1 | **71.1**±0.2 | **57.9**±0.1 |

(b) Batch Size 1

| Methods | S | R | V2 | Avg |
|---|---|---|---|---|
| ResNet50-GN | 29.2 | 40.8 | 68.9 | 46.3 |
| ● Tent | 31.9±0.2 | 42.1±0.1 | 68.9±0.0 | 47.6±0.0 |
| ● CoTTA | 26.7±0.0 | 40.6±0.3 | 68.7±0.2 | 45.3±0.1 |
| ● SAR | 31.6±0.2 | 41.9±0.1 | 68.9±0.1 | 47.5±0.1 |
| ● DeYO | 37.0±0.5 | 46.0±0.2 | **69.0**±0.1 | 50.7±0.3 |
| ● TTE (w. DeYO) | **39.1**±0.1 | **47.1**±0.0 | **69.0**±0.0 | **51.7**±0.1 |
| ViTBase | 18.2 | 43.1 | 66.2 | 37.9 |
| ● Tent | 7.0±0.2 | 40.6±0.5 | 69.1±0.1 | 38.9±0.1 |
| ● CoTTA | 24.0±0.0 | 40.6±0.0 | 66.9±0.0 | 45.6±0.0 |
| ● SAR | 26.4±2.8 | 44.7±0.6 | 69.4±0.2 | 46.8±1.0 |
| ● DeYO | 43.6±0.5 | 59.8±0.3 | 71.2±0.2 | 58.2±0.1 |
| ● TTE (w. DeYO) | **45.0**±0.1 | **61.3**±0.2 | **71.3**±0.1 | **59.2**±0.0 |

# F LIMITATIONS AND FUTURE WORKS

The proposed TTE, as a TTA approach, performs optimization with limited, unlabeled data during inference. In scenarios where test data is identical to training data, over-tuning hyperparameters based on the performance of test data could be controversial. To avoid this risk, we maintained consistent hyperparameter settings across all experiments, although these settings were still determined by performance outcomes. Additionally, reliance on hyperparameter values tailored for specific out-of-distribution characteristics may lead to performance degradation when encountering out-of-distribution scenarios not anticipated in pre-experimental setups. Therefore, developing an effective validation method that determines hyperparameters using either in-distribution data or a subset of out-of-distribution data represents a valuable direction for future research.

