# OpenReview forum: "Test-Time Ensemble via Linear Mode Connectivity: A Path to Better Adaptation"
_ICLR.cc/2025/Conference — ICLR 2025 Poster_

### Official Review · Reviewer_sF3M · 2024-10-26

**Soundness:** 3
**Presentation:** 2
**Contribution:** 2
**Rating:** 6
**Confidence:** 4

**Summary:**

This paper considers a new problem, test-time ensemble (TTE), which aims at using multiple models generated during TTA. This paper first formulates the test-time ensemble problem. The paper also proposes the weight average and dropout as the baseline methods to evaluate the performances.

The contributions can be summarized as:

(1) The author revealed that TTA models exhibit linear mode connectivity, an in- triguing insight that simplifies and enhances the adaptation process.

(2) The author introduced Test-Time Ensemble (TTE), a novel and computationally efficient approach that not only enriches model representations but also stabilize TTA optimization through de-biased and noise- robust knowledge distillation.

(3) TTE integrated effortlessly with existing TTA methods, enhancing adaptation in diverse scenarios and showing potential for applicability to future TTA methods.

**Strengths:**

(1) This paper proposes the new problem, test-time problem, which is different from previous test-time adaptations. I believe this problem has some practical applications.

(2) The paper proposes some simple baseline methods that can effectively address the problem.

**Weaknesses:**

(1) The analysis of test-time adaptation does not inspire the new methods. The moving average ensemble methods are popularly adapted in self-supervised learning and ensemble methods. I consider the Linear Mode Connectivity theory should tell the reason and the situation that the models generated during test-time adaptation.


(2) Limited technical novelty: this paper proposes the two-branch structure and leverage the weight average to improve the performance. Similar techniques are implemented in https://github.com/huggingface/pytorch-image-models. I do not see anything new compared to what have been proved in image classification.

(3) Unclear description. In section 3, the de-biased distillation subsection does not describe clearly where the bias comes from. I suggest the author should explain the bias again. Also, I can not understand what the connection between the spike phenomena of the accuracy curve and the bias.

**Questions:**

Please see the weakness.

---

> ### Author Response · Authors · 2024-11-21
> **Response to Reviewer sF3M (1/2)**
>
> Thank you for recognizing the strengths of our work, including the proposal of new problem and its practical applications. We also appreciate your thoughtful feedback, which highlights areas where further clarification can enhance the paper. Below, we address your comments and questions, aiming to fully resolve your concerns.
>
> > **W1. The analysis of test-time adaptation does not inspire the new methods. The moving average ensemble methods are popularly adapted in self-supervised learning and ensemble methods. I consider the Linear Mode Connectivity theory should tell the reason and the situation that the models generated during test-time adaptation.**
>
> Thank you for raising this concern. We would like to clarify the motivation and methodological novelty of our work and how it builds upon the insight from linear mode connectivity (LMC).
>
> Linear mode connectivity reveals that TTA models, even when adapted to different distributions, can be **weight-averaged across a wide range of combinations** without degrading performance. As shown in **Figure 1**, we empirically validate this property by demonstrating that weight-averaging with TTA models, adapted to distinctly different distributions (noise and blur), consistently maintain or improve performance.
>
> Building on this insight, we propose **adaptive ensemble strategies** that dynamically determine the weight-averaging coefficient (i.e., momentum) to construct ensembles that better leverage the representation diversity of TTA models. This adaptive scheme is particularly valuable in **online TTA**, where data distributions are unknown and subject to dynamic shifts. In **Figure 5**, the Continual TTA experiments demonstrate that our adaptive ensembles significantly outperform static ensembles by effectively incorporating the representation diversity across models.
>
> While some elements of our framework may resemble previous techniques, we provide, for the first time, empirical evidence that the TTA process can effectively leverage ensemble techniques by demonstrating linear model connectivity during adaptation. To the best of our knowledge, this work represents the **first application of linear mode connectivity in TTA research**, introducing a novel adaptive ensemble method tailored to this context. This new perspective on TTA optimization offers valuable benefits to this community.
>
> However, we acknowledge that our initial study did not exhaustively evaluate all potential situations in the TTA process. To address this concern, we have conducted additional analyses to verify the robustness of LMC under more diverse conditions. Building on our previous work with two TTA models, we extended the evaluation to scenarios involving four TTA models and also considered continual TTA conditions. The results provide that ensembles generally maintain or improve performance in these expanded settings, demonstrating that linear mode connectivity remains valid, as detailed below.
>
> ### LMC analysis with four TTA models. Averaged classification accuracy (\%) with ImageNet-C.
> | | Target shifts | Non-target shifts |
> |-|:-:|:-:|
> | No Adapt. | 24.6 | 31.8 |
> | TTA (Gauss) | 37.7 | 47.1 |
> | TTA (Defocus) | 40.3 | 43.2 |
> | TTA (Snow) | 38.7 | 44.0 |
> | TTA (Contrast) | 38.6 | 40.4 |
> | Ensemble (G+D+S+C) | **45.5** | **47.6** |
>
> ### LMC analysis with continual TTA processes. Averaged classification accuracy (\%) with ImageNet-C.
> | | Target shifts | Non-target shifts |
> |-|:-:|:-:|
> | No Adapt. | 24.6 | 31.8 |
> | TTA (G$\rightarrow$D$\rightarrow$S$\rightarrow$C) | 49.4 | 49.6 |
> | Ensemble (G+D+S+C) | **50.7** | **51.5** |
> | TTA (C$\rightarrow$G$\rightarrow$D$\rightarrow$S) | 45.3 | 46.8 |
> | Ensemble (C+G+D+S) | **52.2** | **51.1** |
> | TTA (S$\rightarrow$C$\rightarrow$G$\rightarrow$D) | **54.6** | **52.3** |
> | Ensemble (S+C+G+D) | 53.1 | 51.6 |
> | TTA (D$\rightarrow$S$\rightarrow$C$\rightarrow$G) | 0.2 | 1.6 |
> | Ensemble (D+S+C+G) | **7.4** | **20.0** |
>
> To clarify the motivation of TTE, all the response will be included in the final revised manuscript and the details of the additional experiments will be included in **Table 7 and 8**. We hope this addresses your concerns and highlights the novelty and applicability of our method.
>
> > **W2. Limited technical novelty: this paper proposes the two-branch structure and leverage the weight average to improve the performance. Similar techniques are implemented in https://github.com/huggingface/pytorch-image-models. I do not see anything new compared to what have been proved in image classification.**
>
> Expanding to the previous response to W1, we would like to further emphasize the distinct contributions of TTE, which go beyond conventional methods.

---

> ### Author Response · Authors · 2024-11-21
> **Response to Reviewer sF3M (2/2)**
>
> Our methodological contributions are summarized in two key aspects: 1) adaptive ensemble strategies inspired by linear mode connectivity (Section 3.1) and 2) debiased and noise-robust knowledge distillation as a stable optimization strategy (Section 3.2). The second contributions of TTE address critical challenges in test-time adaptation (TTA). Below, we elaborate on these contributions to clarify their novelty:
>
> **Debiased and noise-robust knowledge distillation**: We propose robust knowledge distillation to address two degradation factors in ensemble representation before distillation: 1) **Bias** as a prominent aspect of model collapse and 2) **Noise** as prediction errors, to ensure the stability of unsupervised TTA optimization. Especially, we would like to emphasize the technical novelty of the **de-biased distillation** approach in addressing the **model collapse issues**, which is a persistent challenging in TTA. Briefly, the de-biasing scheme identifies bias-guiding samples and adjusts their representation by reducing the bias measured during TTA. It induces label smoothing effects and performs **a regularizer for TTA only relying on unsupervised entropy minimization**, thereby achieves stable optimization by mitigating model collapse that often results in near-zero accuracy or catastrophic forgetting. Notably, extensive experiments in this paper (**Table 1-4**) demonstrate that TTE prevents collapse across four benchmark datasets and four test-time scenarios, where baseline methods frequently fail.
>
> For further clarity, we revisited the **continual TTA experiments** in Table 3, a challenging scenario where baseline methods often experience model collapse and catastrophic forgetting. Expanding on prior work with **DeYO+TTE**, we conducted additional experiments using **Tent+TTE** and **SAR+TTE** to validate the effectiveness of the debiasing scheme across different baselines. Additionally, we measured the accuracy on the source data immediately after each adaptation process to assess whether TTE effectively mitigates catastrophic forgetting. The results were consistent with those observed for DeYO, demonstrating that the de-biased distillation reliably prevents performance degradation while addressing critical gaps in existing TTA research.
>
> ### Continual TTA with non-i.i.d. conditions. Average accuracy (\%) with ImageNet-C.
> | | ResNet50-GN | ViTBase |
> |-|:-:|:-:|
> | NoAdapt | 30.6 |29.9|
> | Tent    | 0.6 | 3.9 |
> | Tent+TTE | **44.2(+43.6)** | **58.3(+54.4)** |
> | SAR | 23.0 | 46.0 |
> | SAR+TTE | **44.9(+21.9)** | **60.3(+14.3)** |
> | DeYO | 2.8 | 53.5 |
> | DeYO+TTE | **49.7(+46.9)** | **61.6(+8.1)** |
>
> ### Comparison of preventing catastrophic forgetting on Continual TTA with non i.i.d. conditions. Average accuracy (\%) with clean ImageNet.
> | | ResNet50-GN | ViTBase |
> |-|:-:|:-:|
> | NoAdapt | 80.0 |78.0|
> | Tent    | 10.1 | 9.8 |
> | Tent+TTE | **75.1(+65.0)** | **79.9(+70.1)** |
> | SAR | 56.7 | 76.2 |
> | SAR+TTE | **75.4(+18.7)** | **80.4(+4.2)** |
> | DeYO | 8.1 | 38.4 |
> | DeYO+TTE | **71.3(+63.2)** | **77.9(+39.5)** |
>
> The final manuscript will include these distinctions and the additional experimental results in **Table 11 and Figure 8** to highlight their importance.
>
> > **W3. Unclear description. In section 3, the de-biased distillation subsection does not describe clearly where the bias comes from. I suggest the author should explain the bias again. Also, I can not understand what the connection between the spike phenomena of the accuracy curve and the bias.**
>
> We appreciate your suggestion to clarify and improve Section 3. In the revised manuscript, we have thoroughly revised this section by:
> - **Definition of Bias**: We define the degradation factor in ensemble output.
> - **Explaining the Bias Problem**: We have added a detailed explanation of how biased outputs emerge during TTA and negatively impact adaptation performance.
> - **Detailing the Proposed Distillation Scheme**: We provide a multi-step explanation of the proposed scheme, highlighting how it addresses bias for robust TTA optimization.
>
> Briefly, the bias represents a prominent aspect in model predictions when TTA models collapse. We interpreted the collapsing phenomenon through the lens of linear mode connectivity by monitoring loss surface barrier between two TTA models. By introducing a robust knowledge distillation, TTE prevents model collapse and maintains linear mode connectivity during TTA, ensuring ensemble benefits during optimization. We hope these updates address your concerns and enhance the overall presentation of the manuscript.
>
> We are eager to engage in further discussion and would greatly appreciate any additional suggestions or concerns you might have. Your feedback is invaluable in refining and enhancing our work, and we are committed to addressing any remaining issues.

---

> ### Author Response · Authors · 2024-11-25
> **Kind Reminder: Looking Forward to Your Post-Rebuttal Feedback**
>
> We sincerely thank you once again for taking the time to provide your thoughtful and valuable feedback on our work. Due to the limited duration of the rebuttal phase, we kindly ask if you have any post-rebuttal feedback that could help us further enhance the quality of our paper.
>
> As a summary of our previous response:
> - We clarified how the proposed adaptive ensemble scheme is motivated by our discovery of linear mode connectivity in TTA. Furthermore, we performed additional analyses to validate the linear mode connectivity property across various potential TTA scenarios, reinforcing the motivation behind TTE.
> - We also clarified another technical novelty: the robust knowledge distillation approach. To further demonstrate its effectiveness, we additionally performed the experiments using other baseline models (i.e., Tent+TTE and SAR+TTE).
> - Following the reviewer’s comments, we thoroughly revised the method section of the main manuscript to enhance clarity and improve understanding.
>
> We deeply value your feedback and are confident that it has significantly contributed to improving the quality of this paper and strengthening our work.
>
> Best regards,
>
> The Authors

---

> > ### Comment · Reviewer_sF3M · 2024-11-25
> >
> > Thanks for your detailed response. I have gone through all reviews. Now I think my major concerns are addressed. Therefore, I raise the rating a little bit.

---

> > > ### Author Response · Authors · 2024-11-26
> > > **Thanks to Reviewer sF3M**
> > >
> > > We greatly appreciate your recognition of the core contributions and strengths of our work.
> > >
> > > If you have any further questions or suggestions, please feel free to reach out. We are fully committed to addressing any concerns or providing additional clarifications.
> > >
> > > Thank you once again for your constructive and thoughtful review.
> > >
> > > Best regards,
> > >
> > > The Authors

---

### Official Review · Reviewer_NKQc · 2024-10-28

**Soundness:** 4
**Presentation:** 4
**Contribution:** 3
**Rating:** 8
**Confidence:** 4

**Summary:**

The paper introduces the Test-Time Ensemble (TTE), a method designed for TTA using the theory of weights space ensemble, which can be used on top of different TTA methods. The authors show different results for TTA over corruptions with different baselines, and the method seems to work pretty well. Furthermore, the authors also provided results for continual TTA, which is interesting.

**Strengths:**

The paper is well-written, easy to follow, and detailed. I liked how the authors presented the work and motivated toward the problem. Furthermore, the results are motivating, and the idea seems easy to implement on top of different methods (as demonstrated by the authors), which can be beneficial for the community if the authors also provide the full code for reproducibility.

**Weaknesses:**

Personally, I did not see many problems with the paper, but I would suggest the authors proofread again to avoid problems such as the following typo "with lager and more complex" -> "with larger and more complex" in the introduction or "Adaptvie momentum" ->  "Adaptive momentum."

If the authors work on the following points, it will improve a lot the quality of the work:
- I am not so convinced by section 3.2, DE-BIASED AND NOISE-ROBUST KNOWLEDGE DISTILLATION. Could you clarify this a bit more in this section? And maybe make it more clear in the paper.

-  In Equation 5, there is no hyperparameter to balance the terms. I think it should be included, right?

- For Table 3, with continual TTA, the authors compared all other baselines with TTE with DeYO, but not TTE with other variants as well, and for continual TTA, I don't understand how the performance can still perform well in the direction of the adaptation without degrading too much as we can see in the other approaches (for example DeYO goes from 28.1 to 3.7 and then 7.2), for me it only make sense if you ensemble with zero-shot model as well (or reset the model weights) but if this is the case it should be done for all other baselines as well.

- For Table 4, column V2 seems strange, as almost all results in a) are 68.9, even DeYO and DeYO with TTE. Could you also add more results with other batch sizes? (the batch size can play an important role in different algorithms, which can benefit DeYO and not the others. For instance, I recommend taking a look at the paper "Bag of Tricks for Fully Test-Time Adaptation, IEEE/CVF Winter Conference on Applications of Computer Vision. 2024",  which shows the role of batch size in some of the TTA algorithms.

- I would suggest revisiting some of the baselines for Tab 1. I would also consider a baseline with other methods of the local ensemble as well, such as SWA with TTA, and for Tab 4. I would also add other methods with TTE (maybe in the supp. material).


Rebuttal period: All my points were answered well, and I am quite happy with the effort of the authors during the rebuttal phase.

**Questions:**

Here, I am adding the questions that I find relevant for improving the work quality; some of them were already discussed in the Weaknesses section:

- Could you clarify this a bit more in section 3.2? How is it important for the method?

- Do you think the batch size can impact the results, especially the ones provided for the continual TTA?

Please consider answering the points on the weaknesses as well. Furthermore, I am open to discussion, and I think that the work has a good potential for the community.

Rebuttal period: After carefully reading all the answers provided by the authors, I feel that my questions were answered, and I don't have any additional questions. I am confident to change my decision from "marginally above the acceptance threshold" to "accept, good paper".

---

> ### Author Response · Authors · 2024-11-21
> **Response to Reviewer NKQc (1/4)**
>
> We sincerely thank the reviewer for their thoughtful feedback, positive remarks on our work! We are also confident that addressing your comments will further enhance the quality and impact of our paper.
>
> > **W1. I would suggest the authors proofread again to avoid problems such as the following typo "with lager and more complex" -> "with larger and more complex" in the introduction or "Adaptvie momentum" -> "Adaptive momentum."**
>
> Thank you for pointing these out. We have carefully proofread the manuscript again and corrected all typographical errors, including the ones you noted. These changes ensure better readability and precision.
>
> > **W2. I am not so convinced by section 3.2, DE-BIASED AND NOISE-ROBUST KNOWLEDGE DISTILLATION. Could you clarify this a bit more in this section? And maybe make it more clear in the paper.**
>
> We appreciate your suggestion to clarify and improve Section 3.2. In the revised manuscript, we have thoroughly revised this section by:
>
> - **Definition of Bias and Noise**: We define the two degradation factors in ensemble output: bias and noise.
> - **Explaining the Bias Problem**: We describe how biased outputs arise during TTA, disrupt linear mode connectivity, and degrade adaptation performance.
> - **Detailing the Proposed Distillation Scheme**: We provide a multi-step explanation of the proposed scheme, highlighting how it addresses bias and noise for robust TTA optimization.
>
> As an extension to this clarification, we provide a brief summary of Section 3.2 and address the related reviewer question (**Q1: How important is this method?**). In this section, we propose de-biased and noise-robust knowledge distillation to address two degradation factors in ensemble representation before distillation: 1) **Bias** as a prominent aspect of model collapse and 2) **Noise** as prediction errors. The schemes do not directly improve adaptation capability but ensure **the stability of unsupervised TTA optimization**.
>
> Especially, we would like to emphasize the technical novelty and effectiveness of the **de-biased distillation** approach in addressing the bias issues, which manifests as TTA models begin to collapse. Model collapse is a persistent challenge in TTA, where improperly biased predictions lead to all samples being misclassified into a single class. By reducing prediction bias, the proposed de-biasing scheme induces label smoothing effects and performs as a **regularizer for TTA only relying on unsupervised entropy minimization**. Therefore, TTE mitigates model collapse and stabilizes the adaptation process. Extensive experiments in our paper (**Table 1-4**) demonstrate that TTE consistently prevents collapse across four benchmark datasets and four test-time scenarios, where baseline methods frequently fail. These results underscore the robustness of our approach.
>
> > **W3. In Equation 5, there is no hyperparameter to balance the terms. I think it should be included, right?**
>
> Thank you for this observation. You are correct that a hyperparameter to balance the terms could potentially be optimized to enhance adaptation performance in TTE. However, we chose to weight the two objective terms equally to avoid the challenges associated with hyperparameter selection during test time.
>
> To address potential overfitting concerns, we deliberately avoided over-tuned hyperparameter configurations for specific tasks or benchmarks. Instead, we applied identical hyperparameter settings across all four benchmarks and four test-time scenarios. This approach underscores the generality and adaptability of TTE without reliance on task-specific tuning.
>
> We have clarified this in the revised manuscript with the following explanation: "To avoid over-tuned hyperparameter configurations, the two objective terms are assigned equal weights."

---

> ### Author Response · Authors · 2024-11-21
> **Response to Reviewer NKQc (2/4)**
>
> > **W4. For Table 3, with continual TTA, the authors compared all other baselines with TTE with DeYO, but not TTE with other variants as well, and for continual TTA, I don't understand how the performance can still perform well in the direction of the adaptation without degrading too much as we can see in the other approaches (for example DeYO goes from 28.1 to 3.7 and then 7.2), for me it only make sense if you ensemble with zero-shot model as well (or reset the model weights) but if this is the case it should be done for all other baselines as well.**
>
> We notify that we did not employ any parameter initialization/reset schemes with a pre-trained zero-shot model during TTA processes. Instead, TTE only relies on robust distillation to prevent model collapse (Please refer to the response to W2). The continual TTA with non-i.i.d. conditions (Table 3) is a challenging scenario, where TTA models are prone to collapse. The experiments were introduced to rigorously assess the robustness of TTE, contributed by the proposed knowledge distillation scheme described in the Section 3.2. Notably, the extensive experiments in this paper (Table 1-4) demonstrate that the integration of TTE effectively avoids collapsing with four test-time scenarios and four datasets. To improve clarity, we conducted additional experiments based on the reviewer’s comments. The details of the experiments are as follows:
>
> **Additional experiments with Tent+TTE and SAR+TTE**: To validate the robustness of TTE further, we revisited the experiments in Table 3 and conducted experiments with other baselines integrated with TTE (e.g., **Tent+TTE** and **SAR+TTE**). The results are consistent with the results of DeYO+TTE in Table 3, as follows:
>
> ### Continual TTA with non-i.i.d. conditions. Average accuracy (\%) with ImageNet-C.
> | | ResNet50-GN | ViTBase |
> |-|:-:|:-:|
> | NoAdapt | 30.6 |29.9|
> | Tent    | 0.6 | 3.9 |
> | Tent+TTE | **44.2(+43.6)** | **58.3(+54.4)** |
> | SAR | 23.0 | 46.0 |
> | SAR+TTE | **44.9(+21.9)** | **60.3(+14.3)** |
> | DeYO | 2.8 | 53.5 |
> | DeYO+TTE | **49.7(+46.9)** | **61.6(+8.1)** |
>
> **Additional experiments for catastrophic forgetting**: TTA methods often experience severe performance degradation on in-distribution data after adaptation, a phenomenon known as catastrophic forgetting. Following [1], we concurrently measured the accuracy on clean ImageNet dataset right after each adaptation to a distribution in the above experiments. The results show that the integration with TTE successfully avoids forgetting issues, while other baseline methods suffer from it, as follows.
>
> ### Comparison of preventing catastrophic forgetting on Continual TTA with non i.i.d. conditions. Average accuracy (\%) with clean ImageNet.
> | | ResNet50-GN | ViTBase |
> |-|:-:|:-:|
> | NoAdapt | 80.0 |78.0|
> | Tent    | 10.1 | 9.8 |
> | Tent+TTE | **75.1(+65.0)** | **79.9(+70.1)** |
> | SAR | 56.7 | 76.2 |
> | SAR+TTE | **75.4(+18.7)** | **80.4(+4.2)** |
> | DeYO | 8.1 | 38.4 |
> | DeYO+TTE | **71.3(+63.2)** | **77.9(+39.5)** |
>
> The details of the additional experiments (i.e., accuracy for each shift) have been included in the **Figure 8** and **Table 11** in the final revision.
>
> > W5. For Table 4, column V2 seems strange, as almost all results in a) are 68.9, even DeYO and DeYO with TTE. Could you also add more results with other batch sizes? (the batch size can play an important role in different algorithms, which can benefit DeYO and not the others. For instance, I recommend taking a look at the paper "Bag of Tricks for Fully Test-Time Adaptation, IEEE/CVF Winter Conference on Applications of Computer Vision. 2024", which shows the role of batch size in some of the TTA algorithms.
>
> Thank you for your insightful comments and paper recommendation. We address your comments point by point below.
>
> **Clarification on V2 Results**: ImageNet-V2 consists of data sampled after a decade of progress on the original ImageNet dataset and is used in our work to measure adaptation performance under intrinsic distribution shifts. While conventional TTA methods, including TTE, show promising results under extrinsic shifts (ImageNet-C, R, and S), they fail to provide performance gains for intrinsic shifts. This finding highlights the need for further investigation into handling such shifts effectively.
>
> **Batch Size Effect**: Previous studies [2, 3], including the one you referenced, have demonstrated that batch size significantly impacts TTA models, particularly those using batch normalization. Acknowledging this effect, we aimed to avoid the instability caused by batch normalization in this paper. To achieve this, we employed architectures that are robust to batch size variations, specifically Vision Transformers with layer normalization (ViTBase) and ResNet50 with group normalization (ResNet50-GN). This design choice ensures that TTA methods, even without TTE, exhibit stable adaptation performance under extreme small-batch settings, such as Batch Size 1 (as shown in **Tables 1 and 4**).

---

> ### Author Response · Authors · 2024-11-21
> **Response to Reviewer NKQc (3/4)**
>
> In response to your comments, we revisited the experiments in Table 4, varying the batch sizes to assess their impact. The results confirm that performance improvements achieved by TTE remain consistent across different batch sizes.
>
> ### Averaged classification accuracy (\%) with the change of batch size in natural distribution shifts (ResNet50-GN).
> | | BS1 | BS4 | BS16 | BS64 |
> |-|:-:|:-:|:-:|:-:|
> | NoAdapt | 46.3 | 46.3 | 46.3 | 46.3 |
> | DeYO    | 50.6 | 50.6 | 50.8 | 49.2 |
> | DeYO+TTE | **51.7(+1.1)** | **51.9(+1.3)** | **51.8(+1.0)** | **50.5(+1.3)** |
>
> ### Averaged classification accuracy (\%) with the change of batch size in natural distribution shifts (VitBase).
> | | BS1 | BS4 | BS16 | BS64 |
> |-|:-:|:-:|:-:|:-:|
> | NoAdapt | 37.9 | 37.9 | 37.9 | 37.9 |
> | DeYO    | 58.2 | 58.2 | 58.4 | 58.1 |
> | DeYO+TTE | **59.2(+1.0)** | **59.1(+0.9)** | **59.2(+0.8)** | **58.6(+0.5)** |
>
> In the revised main manuscript, we have clarified the rationale behind selecting these architectures and included a reference to the recommended paper, which provides a solid foundation for our design choices, as follows: "Architectures with batch normalization were excluded due to their batch size sensitivity and instability during the TTA process (Niu et al., 2023; Mounsaveng et al., 2024)."
>
> > **W6. I would suggest revisiting some of the baselines for Tab 1. I would also consider a baseline with other methods of the local ensemble as well, such as SWA with TTA, and for Tab 4. I would also add other methods with TTE (maybe in the supp. material).**
>
> Thank you for your insightful suggestions to enhance the quality of our paper. Following your comments, we conducted additional experiments to address these points:
>
> **Comparison with Stochastic Weight Averaging (SWA)**: To evaluate the effectiveness of the proposed adaptive weight-averaging method, we compared it with stochastic weight averaging (SWA) [4], which inspired TTE as one of offline generalization research. For this comparison, we created a TTE variant by replacing the proposed ensemble strategy with SWA while keeping all other components unchanged. SWA employs uniform averaging of SGD iterates (generated TTA models) and is implemented as: $w_{\text{swa}}\gets \frac{w_{\text{swa}}\cdot n_{\text{models}}+w}{n_{\text{models}}+1}$. In response to reviewer comments, experiments were conducted using the scenario from Table 1, specifically the ImageNet-C under the Label Shifts setup. The results demonstrate that the proposed adaptive averaging scheme outperforms SWA, highlighting its adaptive mechanism advantages for online TTA compared to SWA's uniform averaging. The results are summarized as follows:
>
> ### Comparison study between different weight averaging schemes. Averaged accuracy with ImageNet-C (Label Shifts).
> | | ResNet50-GN | ViTBase |
> |-|:-:|:-:|
> | NoAdapt | 30.6 |29.9|
> | DeYO    | 40.8 | 61.3 |
> | +TTE (SWA) | 47.4(+7.4) | 64.7(+3.4) |
> | +TTE (Ours) | **50.7(+9.9)** | **65.2(+3.9)** |
>
> **Additional Baselines with TTE**: We revisited the experiments in Table 4 and included results for **Tent+TTE** and **SAR+TTE**. These experiments further validate the effectiveness of TTE, with results consistent with those observed with DeYO+TTE. The performance is summarized as follows:
>
> ### Averaged classification accuracy (%) with natural distribution shifts (Label Shifts).
> | | ResNet50-GN | ViTBase |
> |-|:-:|:-:|
> | NoAdapt | 46.3 | 37.9 |
> | Tent    | 47.1 | 39.8 |
> | Tent+TTE | **48.1(+1.0)** | **52.7(+12.9)** |
> | SAR | 47.0 | 43.8 |
> | SAR+TTE | **47.8(+0.8)** | **52.6(+8.8)** |
> | DeYO | 49.3 | 57.3 |
> | DeYO+TTE | **50.3(+1.0)** | **57.9(+0.6)** |
>
> ### Averaged classification accuracy (%) with natural distribution shifts (Batch Size 1).
> | | ResNet50-GN | ViTBase |
> |-|:-:|:-:|
> | NoAdapt | 46.3 | 37.9 |
> | Tent    | 47.6 | 38.9 |
> | Tent+TTE | **49.7(+2.1)** | **53.8(+14.9)** |
> | SAR | 47.5 | 46.8 |
> | SAR+TTE | 47.5(+0.0) | **51.0(+4.2)** |
> | DeYO | 50.7 | 58.2 |
> | DeYO+TTE | **51.7(+1.0)** | **59.2(1.0)** |
>
> The additional experiments and their details have been included in **Table 12 and 13** of the revised manuscript.
>
> > **Q1. Could you clarify this a bit more in section 3.2? How is it important for the method?**
>
> Please refer to the response to W2.
>
> > **Q2. Do you think the batch size can impact the results, especially the ones provided for the continual TTA?**
>
> Please refer to the response to W5.
>
> All responses will be included in the final revision. We are always open to further discussion, so if you have any additional concerns or suggestions, please do not hesitate to share them—we greatly value your feedback and are committed to improving the quality of our work.

---

> ### Author Response · Authors · 2024-11-21
> **Response to Reviewer NKQc (4/4)**
>
> > References
> > 1. Shuaicheng Niu, Jiaxiang Wu, Yifan Zhang, Yaofo Chen, Shijian Zheng, Peilin Zhao, and Mingkui Tan. Efficient test-time model adaptation without forgetting. In International Conference on Machine Learning, pp. 16888–16905. PMLR, 2022.
> > 2. Shuaicheng Niu, Jiaxiang Wu, Yifan Zhang, Zhiquan Wen, Yaofo Chen, Peilin Zhao, and
> Mingkui Tan. Towards stable test-time adaptation in dynamic wild world. arXiv preprint
> arXiv:2302.12400, 2023.
> > 3. Taesik Gong, Jongheon Jeong, Taewon Kim, Yewon Kim, Jinwoo Shin, and Sung-Ju Lee. Note: Ro-bust continual test-time adaptation against temporal correlation. Advances in Neural Information Processing Systems, 35:27253–27266, 2022.
> > 4. Pavel Izmailov, Dmitrii Podoprikhin, Timur Garipov, Dmitry Vetrov, and Andrew Gordon Wil-
> son. Averaging weights leads to wider optima and better generalization. arXiv preprint
> arXiv:1803.05407, 2018.

---

> > ### Comment · Reviewer_NKQc · 2024-11-23
> >
> > Hello, I would like to thank the authors for their detailed answers and additional clarifications about the work. I think that all my questions were addressed, and I am comfortable with the answers. Thus, I am changing my score from  "marginally above the acceptance threshold" to "accept, good paper".

---

> ### Author Response · Authors · 2024-11-23
> **Thanks to Reviewer NKQc**
>
> We sincerely appreciate your recognition and acknowledgment of the strengths of our work.
>
> Should you have any additional questions or comments, please do not hesitate to share them with us. We are fully committed to addressing them in detail.
>
> Thank you again for your constructive and thoughtful review.
>
> Best regards,
>
> The Authors

---

### Official Review · Reviewer_xGvc · 2024-11-07

**Soundness:** 3
**Presentation:** 3
**Contribution:** 2
**Rating:** 6
**Confidence:** 4

**Summary:**

The paper proposes a method called Test-Time Ensemble (TTE), which uses an ensemble strategy to dynamically enrich model representations during online test-time adaptation (TTA). TTE constructs an ensemble network by averaging the parameter weights of different TTA models, which are continuously updated using test data. This weight averaging technology captures model diversity and improves representation quality without increasing the computational burden of managing multiple models. TTE further combines dropout to promote diverse collaboration of representations within TTA models, and also proposes a debiased and noise-resistant knowledge distillation scheme to stabilize the learning of TTA models in the ensemble. TTE can be seamlessly integrated with existing TTA methods, enhancing their adaptive capabilities in various challenging scenarios.

**Strengths:**

1. TTE utilizes an ensemble strategy to dynamically enrich model representations during online test-time adaptation (TTA), which is an interesting approach.
2. TTE constructs an ensemble network by averaging the parameter weights of different TTA models, and this weight averaging captures model diversity, improving representation quality without increasing the computational burden of managing multiple models.
3. TTE further promotes the diversity of representations within TTA models by combining with dropout, and proposes a debiased and noise-resistant knowledge distillation scheme to stabilize the learning of TTA models in the ensemble.
4. The experiments are extensive and the results are superior to other compared methods.

**Weaknesses:**

1. The TTE method involves multiple hyperparameters, such as the momentum coefficient , dropout ratio and temperature, which may affect the stability and generalization of the method. Further research on how to reduce the dependence on hyperparameters is crucial.
2. TTE integrates many well-established techniques such as ensemble, dropout, knowledge distillation, which have been utilized in TTA or few shot learning. The combination has weaken the novelty of the paper, and the unique contributions of the paper should be classified.
3. How to conduct the weight-space ensemble without adding computational complexity? Will the technique increase storage consumption?
4. The knowledge distillation-based debiasing and anti-noise strategies proposed in the paper may not be able to completely solve the problem of noisy test data. How to solve the scenario that the pseudo labels are incorrect?

**Questions:**

1. Why the results of CoTTA in Continual TTA with non-i.i.d. conditions are only 2.2% and 3.4%?

---

> ### Author Response · Authors · 2024-11-21
> **Response to Reviewer xGvc (1/2)**
>
> We thank the reviewer for their thoughtful feedback and for recognizing the strengths of our work. Below, we address the concerns raised and clarify our contributions.
>
> > **W1. The TTE method involves multiple hyperparameters, such as the momentum coefficient , dropout ratio and temperature, which may affect the stability and generalization of the method. Further research on how to reduce the dependence on hyperparameters is crucial.**
>
> Thanks for the reviewer highlighting the importance of hyperparameter sensitivity and its impact on the stability and generalization of TTE. We address this concern as follows:
>
> - **Demonstrated low hyperparameter sensitivity**: We would like to emphasize that TTE demonstrates low sensitivity to hyperparameters. As shown in **Figures 6 and 7** of the paper, TTE consistently outperforms the baseline across a range of hyperparameter settings. These results indicate that TTE maintains effectiveness even with approximately chosen hyperparameters.
>
> - **Uniform hyperparameter configuration**: To further address potential overfitting concerns, we intentionally avoided over-tuned hyperparameter configuration for specific tasks or benchmarks. Instead, we applied identical hyperparameter settings across all four benchmarks and all four test-time scenarios. This generality highlights the adaptability of TTE without reliance on tuning.
>
> - **Limitations**: Nevertheless, we recognize the reviewer’s valid point regarding the challenge of hyperparameter selection in test time. To reflect this, we have added a limitation section to the paper (**Appendix F**), explicitly discussing the difficulty of hyperparameter settings.
>
> The final manuscript will incorporate all the responses. We hope this demonstrates our commitment to addressing hyperparameter concerns and enhancing the practical applicability of TTE.
>
> > **W2. TTE integrates many well-established techniques such as ensemble, dropout, knowledge distillation, which have been utilized in TTA or few shot learning. The combination has weaken the novelty of the paper, and the unique contributions of the paper should be classified.**
>
> We would like to clarify the novelty of TTE through this response. Our methodological contributions are summarized in two key aspects: 1) adaptive ensemble strategies inspired by linear mode connectivity (Section 3.1) and 2) debiased and noise-robust knowledge distillation as a stable optimization strategy (Section 3.2). To address the reviewer concern, we classify the two technical novelty in TTE, which obviously distinguishes it from conventional methods.
>
> **1. Adaptive ensemble strategies**: TTE introduces adaptive ensemble strategies inspired by the linear mode connectivity property, which ensures the wide range of weight combinations from different models (**Figure 1**). Unlike static ensembling, our approach actively performs ensembling, particularly when TTA models are adapted to new distributions and their representation diverge.  This adaptive mechanism is especially advantageous in **online TTA**, where incoming data distributions are unknown and may shift dynamically. As demonstrated in the Continual TTA experiments (**Figure 5**), these adaptive ensembles significantly outperform static ensembles by effectively leveraging representation diversity.
>
> While some elements of our framework may resemble previous techniques, we provide, for the first time, empirical evidence that the TTA process can effectively leverage ensemble techniques by demonstrating linear model connectivity during adaptation. To the best of our knowledge, this work represents the **first application of linear mode connectivity in TTA research**, introducing a novel adaptive ensemble method specifically tailored to this context. This new perspective on TTA optimization offers valuable benefits to this community.
>
> **2. De-biased distillation**: Unlike conventional knowledge distillation, our proposed de-biased distillation introduces a de-biasing mechanism to refine the ensemble representation before distillation. Practically, the de-biasing scheme identifies bias-guiding samples and adjusts their representation by reducing the bias measured during TTA process. It induces label smoothing effects and performs a **regularizer during TTA only relying on unsupervised entropy minimization**. As a result, TTE achieves stable optimization by 1) maintaining linear mode connectivity between TTA models and 2) mitigating model collapse, a persistent issue in TTA methods (often results in near-zero accuracy or catastrophic forgetting). Notably, extensive experiments in this paper (**Table 1-4**) demonstrate that TTE prevents collapse across four benchmark datasets and four test-time scenarios, where baseline methods frequently fail.

---

> ### Author Response · Authors · 2024-11-21
> **Response to Reviewer xGvc (2/2)**
>
> To further verify the technical novelty of the de-biasing scheme, we revisited the experiments in Table 3 and conducted with **Tent+TTE** and **SAR+TTE**. The scenario in Table 3 is challenging where conventional TTA models frequently suffer from collapse. The results are consistent with the original results of **DeYO+TTE** as follows.
>
> ### Continual TTA with non-i.i.d. conditions. Average accuracy (\%) with ImageNet-C.
> | | ResNet50-GN | ViTBase |
> |-|:-:|:-:|
> | NoAdapt | 30.6 |29.9|
> | Tent    | 0.6 | 3.9 |
> | Tent+TTE | **44.2(+43.6)** | **58.3(+54.4)** |
> | SAR | 23.0 | 46.0 |
> | SAR+TTE | **44.9(+21.9)** | **60.3(+14.3)** |
> | DeYO | 2.8 | 53.5 |
> | DeYO+TTE | **49.7(+46.9)** | **61.6(+8.1)** |
>
> The final manuscript will clarify these contributions to address the reviewer concerns.
>
> > **W3. How to conduct the weight-space ensemble without adding computational complexity? Will the technique increase storage consumption?**
>
> We appreciate the reviewer’s comment and agree that our original expression was too strong. We have revised the manuscript to state: “reducing the computational burden of multiple model inference.” in the Introduction section.
>
> > **W4. The knowledge distillation-based debiasing and anti-noise strategies proposed in the paper may not be able to completely solve the problem of noisy test data. How to solve the scenario that the pseudo labels are incorrect?**
>
> We agree with the reviewer’s comment that the proposed noise-robust distillation cannot completely prevent noisy pseudo label issues. However, we have both empirically and theoretically demonstrated that it effectively alleviates their impact in this paper. This is achieved through the use of reverse KL divergence, which reverses the order of the student and teacher representations. Mathematically, unlike standard KL divergence, the gradient of reverse KL divergence does not depend on the magnitude ratio of student and teacher predictions, making it less sensitive to noisy teacher predictions. These findings align with observations in supervised learning with noisy labels [1]. A detailed mathematical analysis of this noise-robust distillation approach is provided in **Appendix D.2**.
>
> > **Q1. Why the results of CoTTA in Continual TTA with non-i.i.d. conditions are only 2.2\% and 3.4\%?**
>
> CoTTA was originally developed for continual TTA applications. However, it does not account for the challenging scenarios of non-i.i.d. conditions, both in terms of distributions and classes. In this paper, we introduced such scenarios to evaluate the robustness of TTE, as these conditions make conventional TTA models particularly prone to collapse. The observed collapse in CoTTA under another challenging scenarios is consistent with findings reported in previous research [2].
>
> The revised manuscript will incorporate all the responses. We deeply value your feedback and are eager to engage in further discussion. If you have any additional concerns or suggestions, please do not hesitate to let us know.
>
> > References
> > 1. Yisen Wang, Xingjun Ma, Zaiyi Chen, Yuan Luo, Jinfeng Yi, and James Bailey. Symmetric cross entropy for robust learning with noisy labels. In Proceedings of the IEEE/CVF international conference on computer vision, pp. 322–330, 2019b.
> > 2. Longhui Yuan, Binhui Xie, and Shuang Li. Robust test-time adaptation in dynamic scenarios. In Proceedings of the IEEE/CVF Conference on Computer Vision and Pattern Recognition, pp. 15922–15932, 2023.

---

> ### Author Response · Authors · 2024-11-25
> **Kind Reminder: Looking Forward to Your Post-Rebuttal Feedback**
>
> We sincerely thank you once again for taking the time to provide your thoughtful and valuable feedback on our work. Due to the limited duration of the rebuttal phase, we kindly ask if you have any post-rebuttal feedback that could help us further enhance the quality of our paper.
>
> Summary of the previous responses:
> - The low hyperparameter sensitivity of TTE was clarified, and a discussion of the challenges associated with hyperparameter tuning during test-time was included as a limitation in response to the reviewer’s concern.
> - The two key technical contributions of TTE were further clarified, and the manuscript was thoroughly revised to emphasize its distinct contributions.
> - Additional experiments were conducted with other baseline models (Tent and SAR) to further validate the stability of TTE.
> - The discussion of computational costs has been softened to provide a more balanced perspective.
> - The proposed knowledge distillation was clarified to explain how it alleiviates the impacts of noisy predictions.
>
> We deeply value your feedback and are confident that it has significantly contributed to improving the quality of this paper and strengthening our work.
>
> Best regards,
>
> The Authors

---

### Official Review · Reviewer_drM4 · 2024-11-08

**Soundness:** 3
**Presentation:** 4
**Contribution:** 3
**Rating:** 6
**Confidence:** 3

**Summary:**

This paper introduces a novel test-time ensemble approach that can be seamlessly integrated with existing TTA models to enhance adaptation. Specifically, the proposed framework reduces domain gaps through two ensemble strategies: weight averaging of TTA models and dropout. Additionally, a knowledge distillation strategy is employed to mitigate both noise and bias for improving model robustness under different distribution shifts.
Extensive experiments are conducted in different TTA scenarios to demonstrate the superiority of the proposed method over existing baselines.

**Strengths:**

1. Writing quality is good. The paper is well-structured, and clearly written.
2. Good insights. This paper explores TTA as a domain generalization problem, uncovering linear connectivity within TTA models. This perspective suggests that domain generalization techniques could enhance model representations for TTA tasks.
3. SOTA performance. The proposed method achieves the state-of-the-art performance via the integration with different TTA models in various scenarios.
4. Ablations. Ablation experiments are provided to verify the effectiveness of the proposed modules.

**Weaknesses:**

1. Although the results presented in Tables 3 show the performance improvement achieved by the proposed framework in the continual TTA scenario, it is unclear how the method enhances baseline performance in later adaptation stages. Additionally, I would like to know if the proposed method addresses the issue of catastrophic forgetting in this context.

Additional question:
Is it possible to extend the proposed benchmark construction method to dense prediction tasks, such as semantic segmentation? It would be very meaningful if it can be applied to various tasks.

**Questions:**

Please refer to the weaknesses above.

---

> ### Author Response · Authors · 2024-11-21
> **Response to Reviewer drM4**
>
> We sincerely thank the reviewer for their constructive feedback and for recognizing the strengths of our work. We hope to address your comments effectively and further improve the quality of this paper.
>
> > **W1. Although the results presented in Table 3 show the performance improvement achieved by the proposed framework in the continual TTA scenario, it is unclear how the method enhances baseline performance in later adaptation stages. Additionally, I would like to know if the proposed method addresses the issue of catastrophic forgetting in this context.**
>
> The technical contributions of TTE can be summarized in two key aspects: 1) introducing adaptive ensemble strategies from the perspective of linear mode connectivity (Section 3.1), and 2) incorporating debiased and noise-robust knowledge distillation (KD) as a stable optimization strategy during TTA (Section 3.2). The experiments in Table 3, which had a challenging scenario where TTA models are prone to collapse, were introduced to evaluate the **stable optimization in TTE**, induced by the second contribution.
>
> Briefly, the proposed distillation method not only maintains linear mode connectivity during TTA but also effectively alleviates model collapse, a common issue that conventional methods still suffer from (resulting in near-zero accuracy in Table 3 or catastrophic forgetting). Notably, the extensive experiments (Table 1-4) demonstrate that the integration of TTE consistently avoids collapsing across four benchmark datasets and four test-time scenarios.
>
> To improve clarity, we have thoroughly revised the sections discussing the contributions and conducted additional experiments based on the reviewer’s comments. The details of our revisions are as follows:
>
> **Additional experiments for robustness**: To validate the robustness of TTE further, we revisited the experiments in Table 3 and conducted experiments with other baselines integrated with TTE (e.g., **Tent+TTE** and **SAR+TTE**). The results were consistent with the results of **DeYO+TTE** in Table 3, as follows:
>
> ### Continual TTA with non-i.i.d. conditions. Average accuracy (%) with ImageNet-C.
> | | ResNet50-GN | ViTBase |
> |-|:-:|:-:|
> | NoAdapt | 30.6 |29.9|
> | Tent    | 0.6 | 3.9 |
> | Tent+TTE | **44.2(+43.6)** | **58.3(+54.4)** |
> | SAR | 23.0 | 46.0 |
> | SAR+TTE | **44.9(+21.9)** | **60.3(+14.3)** |
> | DeYO | 2.8 | 53.5 |
> | DeYO+TTE | **49.7(+46.9)** | **61.6(+8.1)** |
>
> **Additional experiments for catastrophic forgetting**: Following your comments, we concurrently measured the accuracy on clean ImageNet dataset immediately after each adaptation process in the above experiments. The results show that the integration with TTE successfully avoids forgetting issues, while other baseline methods often suffer from it, as follows.
>
> ### Comparison of preventing catastrophic forgetting on Continual TTA with non i.i.d. conditions. Average accuracy (%) with clean ImageNet..
> | | ResNet50-GN | ViTBase |
> |-|:-:|:-:|
> | NoAdapt | 80.0 |78.0|
> | Tent    | 10.1 | 9.8 |
> | Tent+TTE | **75.1(+65.0)** | **79.9(+70.1)** |
> | SAR | 56.7 | 76.2 |
> | SAR+TTE | **75.4(+18.7)** | **80.4(+4.2)** |
> | DeYO | 8.1 | 38.4 |
> | DeYO+TTE | **71.3(+63.2)** | **77.9(+39.5)** |
>
> **Writing revision**: We have clarified the experimental results in **Section: Continual TTA with non-i.i.d. conditions** for Table 3 and rewritten **Section 3.2** to provide a clearer explanation of the robust knowledge distillation approach. The details of the additional experiments have been included in **Figure 8** and **Table 12** in the final revision.
>
> > **W2. Is it possible to extend the proposed benchmark construction method to dense prediction tasks, such as semantic segmentation? It would be very meaningful if it can be applied to various tasks.**
>
> We appreciate the reviewer’s insightful suggestion regarding the applicability of TTE. Importantly, the proposed TTE framework does not rely on strategies specific to image classification, which suggests that expanding its applicability to dense prediction tasks should be feasible. While we cannot provide experimental results at this time, we are actively preparing experiments to explore this direction and will include them if feasible within the rebuttal period.
>
> We are open to further discussion and welcome any additional concerns or suggestions you may have. Please feel free to share your thoughts, as we are committed to improving the quality of our work.

---

> ### Author Response · Authors · 2024-11-25
> **Response to Reviewer drM4 (Dense Prediction Task)**
>
> Following the reviewer's comments, we extended our experiments to semantic segmentation, a dense prediction task. Using the ViTBase architecture and a pretrained model from DINO v2 [1], we added a linear layer to create a simple baseline model to perform semantic segmentation. This model was fine-tuned on the COCO training set [2] for 10 epochs, achieving 56.0 mIoU on the original COCO validation set.
>
> To simulate distribution shifts, inspired by [3], we reconstructed the COCO validation data with four representative shifts: noise, blur, brightness and JPEG. Using this shifted dataset, we conducted adaptation experiments with Tent and Tent+TTE to evaluate their performance. In this setting, we found that the conventional TTA approach (Tent) showed limited improvement and, in some cases, even degraded performance compared to the baseline model, and as a result, the integration of TTE also showed marginal benefits. We believe this exploration shows the unique challenges of TTA under dense prediction and will promote further research to address these challenges for this community. The detailed results are presented below.
>
> ### Additional TTA experimental results for semantic segmentation with mIOU
> | | Noise | Blur | Brightness | JPEG |
> |-|:-:|:-:|:-:|:-:|
> | NoAdapt | 39.2 |55.4| 53.1 | 35.3 |
> | Tent    | 39.4 | 55.4 | 53.1 | 34.4 |
> | Tent+TTE | 39.5 | 55.5 | 53.1 | 35.3 |
>
> We sincerely thank you once again for taking the time to provide your thoughtful and valuable feedback on our work. Due to the limited duration of the rebuttal phase, we kindly ask if you have any post-rebuttal feedback that could help us further enhance the quality of our paper.
>
> > reference
> > 1. Oquab, Maxime, et al. DINOv2: Learning Robust Visual Features without Supervision, TMLR 2024.
> > 2. Lin, Tsung-Yi, et al. Microsoft coco: Common objects in context. ECCV 2014.
> > 3. Hendrycks, Dan, and Thomas Dietterich. Benchmarking neural network robustness to common corruptions and perturbations. ICLR 2019.

---

> > ### Comment · Reviewer_drM4 · 2024-11-25
> >
> > Thank you very much to the authors for addressing my questions and resolving my concerns. Based on the current discussions, I have decided to maintain a positive rating.

---

> > > ### Author Response · Authors · 2024-11-26
> > > **Thanks to Reviewer drM4**
> > >
> > > We greatly appreciate your recognition of the core contributions and strengths of our work.
> > >
> > > If you have any further questions or suggestions, please feel free to reach out. We are fully committed to addressing any concerns or providing additional clarifications.
> > >
> > > Thank you once again for your constructive and thoughtful review.
> > >
> > > Best regards,
> > >
> > > The Authors

---

### Author Response · Authors · 2024-11-23
**General Response**

We thank the reviewers for their thoughtful feedback and valuable insights. Your comments have helped us identify areas to clarify and strengthen our work. Below, we summarize the key strengths of our contribution, recognized by reviewers.

### **Summarized contributions recognized by the reviewers.**

- The problem formulation, **exploring TTA as a domain generalization problem**, is well-motivated [drM4, NKQc] and is initially proposed in this paper [sF3M].

- **Uncovering linear mode connectivity within TTA models** provides valuable insights [drM4] and enables the development of an **adaptive ensemble strategy** for improving model representations during online TTA [drM4, xGvc].

- TTE also introduces a **debiased and noise-robust knowledge distillation scheme** that stabilizes TTA optimization [xGvc].

- TTE achieves **state-of-the-art (SOTA) performance**, surpassing conventional TTA models [drM4, xGvc], supported by well-motivated experimental results [NKQc].

- **Extensive ablation studies** validate the effectiveness of the proposed modules and highlight their contributions to the overall performance [drM4, xGvc].

- TTE is **practical, easy to implement with baseline models** [NKQc, sF3M], and beneficial to the research community [NKQc].

### **Major revisions in the updated manuscript**

Following the reviewers' suggestions, we have thoroughly revised the paper and uploaded the updated version, which we believe has significantly enhanced the quality of this paper. Here, we would like to highlight the major revisions, as follows.

#### **Introduction**
- The introduction has been revised to clarify the concept of TTE as an adaptive weight ensemble scheme.
- The discussion of computational costs has been softened to provide a more balanced perspective.

#### **Preliminaries**
- Additional analysis has been conducted to validate the property of linear mode connectivity under various potential TTA scenarios, further supporting the motivation for TTE.

#### **Methods**
- The distinct contributions of adaptive ensemble strategies and their significance in online TTA have been clarified and are now more prominently presented in the revised paper.
- Section 3.2 has been revised to improve the clarity and understanding of the proposed knowledge distillation method.

#### **Experiments**
- Additional experiments were conducted with other baseline models (Tent[1]+TTE and SAR[2]+TTE). We revisited the experiments in Tables 3 and 4 to further validate the effectiveness of TTE.
- Additional experiments under continual TTA with non-i.i.d. conditions demonstrated the stability of TTE and its ability to prevent catastrophic forgetting.
- Additional comparison studies with stochastic weight averaging [3], a prior method for offline domain generalization, were performed to highlight the advantages of TTE.

Furthermore, we will incorporate all the responses provided during the discussion phase into the final manuscript. For details, please refer to our responses to each reviewer’s comments.

> References
> 1. Dequan Wang, Evan Shelhamer, Shaoteng Liu, Bruno Olshausen, and Trevor Darrell. Tent: Fully test-time adaptation by entropy minimization. International Conference on Learning Representations (ICLR) 2020.
> 2. Shuaicheng Niu, Jiaxiang Wu, Yifan Zhang, Zhiquan Wen, Yaofo Chen, Peilin Zhao, and Mingkui Tan. Towards stable test-time adaptation in dynamic wild world. International Conference on Learning Representations (ICLR) 2023.
> 3. Pavel Izmailov, Dmitrii Podoprikhin, Timur Garipov, Dmitry Vetrov, and Andrew Gordon Wil- son. Averaging weights leads to wider optima and better generalization. Conference on Uncertainty in Artificial Intelligence (UAI) 2018.

---

### Meta-Review · Area_Chair_dAXu · 2024-12-17

**Metareview:**

The paper proposes a method called Test-Time Ensemble (TTE), which enriches model representations during online test-time adaptation (TTA) inspired by domain generalization. The proposed method is well-motivated and technically solid. The experimental results are extensive and convincing. The authors successfully addressed the major concerns raised by the reviewers about the technical details, novelty, possible extension, experiments, and writing during the rebuttal. All reviewers agree to accept this paper. Overall, it is a technically solid paper that meets the expectations of the ICLR community.

**Additional Comments On Reviewer Discussion:**

The major concerns raised by the reviewers include technical details, novelty, possible extension, experiments, and writing. The rebuttal successfully addresses these issues. At the end of the discussion, all reviewers agree to accept this paper.

---

### Decision · Program_Chairs · 2025-01-22

Accept (Poster)